# Aging impairs cold-induced beige adipogenesis and adipocyte metabolic reprogramming

Corey D Holman[1,2], Alexander P Sakers[1,2], Ryan P Calhoun[1,2], Lan Cheng[1,2], Ethan C Fein[1,2], Christopher Jacobs[3,4], Linus Tsai[3,4,5], Evan D Rosen[3,4,5], Patrick Seale[1,2]*

[1]Institute for Diabetes, Obesity & Metabolism, Perelman School of Medicine at the University of Pennsylvania, Philadelphia, United States; [2]Department of Cell and Developmental Biology; Perelman School of Medicine at the University of Pennsylvania, Philadelphia, United States; [3]Division of Endocrinology, Diabetes, and Metabolism, Beth Israel Deaconess Medical Center, Boston, United States; [4]Broad Institute of MIT and Harvard, Cambridge, United States; [5]Harvard Medical School, Boston, United States

*For correspondence:
sealep@pennmedicine.upenn.
edu

Competing interest: The authors declare that no competing interests exist.

**Abstract** The energy-burning capability of beige adipose tissue is a potential therapeutic tool for reducing obesity and metabolic disease, but this capacity is decreased by aging. Here, we evaluate the impact of aging on the profile and activity of adipocyte stem and progenitor cells (ASPCs) and adipocytes during the beiging process in mice. We found that aging increases the expression of *Cd9* and other fibro-inflammatory genes in fibroblastic ASPCs and blocks their differentiation into beige adipocytes. Fibroblastic ASPC populations from young and aged mice were equally competent for beige differentiation in vitro, suggesting that environmental factors suppress adipogenesis in vivo. Examination of adipocytes by single nucleus RNA-sequencing identified compositional and transcriptional differences in adipocyte populations with aging and cold exposure. Notably, cold exposure induced an adipocyte population expressing high levels of de novo lipogenesis (DNL) genes, and this response was severely blunted in aged animals. We further identified *Npr3*, which encodes the natriuretic peptide clearance receptor, as a marker gene for a subset of white adipocytes and an aging-upregulated gene in adipocytes. In summary, this study indicates that aging blocks beige adipogenesis and dysregulates adipocyte responses to cold exposure and provides a resource for identifying cold and aging-regulated pathways in adipose tissue.

## eLife assessment

This **fundamental** study provides evidence that de novo beige adipogenesis from Pdgfra+ adipocyte progenitor cells is blocked during early aging in subcutaneous fat. The depth of the data at early ages is **compelling**, with rigorous cell tracing methodology employed. The study will aid in identifying new approaches to switch dormant adipocytes into an active thermogenic phenotype, and should be of interest to cell biologists at large.

## Introduction

Brown and beige fat cells are specialized to burn calories for heat production and have the capacity to reduce obesity and metabolic disease. Brown adipocytes are localized in dedicated brown adipose tissue (BAT) depots, whereas beige adipocytes develop in white adipose tissue (WAT) in response

to cold exposure, and other stimuli (*Wang and Seale, 2016*). Adult humans possess thermogenic adipose depots that appear to resemble rodent beige adipose tissue (*Wu et al., 2012*; *Jespersen et al., 2013*). Brown and beige adipocytes share similar cellular features such as abundant mitochondria, multilocular lipid droplets, and expression of thermogenic genes like Uncoupling Protein-1 (UCP1). UCP1, when activated, dissipates the mitochondrial proton gradient, leading to high levels of substrate oxidation and heat production (*Cannon and Nedergaard, 2004*). Brown and beige adipocytes can also produce heat via UCP1-independent futile cycles (*Chouchani et al., 2019*).

Increasing beige fat development in mice reduces obesity and improves insulin sensitivity, whereas ablation of beige fat in mice causes metabolic dysfunction (*Cederberg et al., 2001*; *Seale et al., 2011*; *Cohen et al., 2014*; *Shao et al., 2016*; *Stine et al., 2016*). Furthermore, transplantation of human beige adipocytes into obese mice reduces liver steatosis and improves metabolic health (*Shao et al., 2016*). Beige adipocytes develop via the de novo differentiation of adipocyte stem and progenitor cells (ASPCs) or through induction of the thermogenic program in adipocytes (*Shao et al., 2019*; *Ferrero et al., 2020*; *Sakers et al., 2022*).

Human and mouse thermogenic adipose tissue activity declines with aging, predisposing to cardiometabolic disease and limiting the potential of brown/beige fat-targeted therapies (*Yoneshiro et al., 2011*; *Cypess et al., 2012*; *Rogers et al., 2012*; *Berry et al., 2017*; *Wang et al., 2019*; *Becher et al., 2021*). In mice, beige adipose tissue is reduced by 'middle-age' (i.e. 1-year-old), preceding many of the damaging effects of old age on organ function (*Rogers et al., 2012*; *Berry et al., 2017*; *Gonçalves et al., 2017*). The aging-associated decline in beige fat activity can occur independently of increases in body weight (*St Onge, 2005*; *Rogers et al., 2012*). A variety of processes and pathways have been linked to the aging-induced deficit in beige fat formation, including diminished proliferation and cellular senescence of ASPCs (*Berry et al., 2017*), increased fibrosis (*Wang et al., 2019*), increased inflammation (*Ghosh et al., 2019*), accumulation of anti-adipogenic regulatory cells (*Nguyen et al., 2021*), and reduced adrenergic tone (*Rogers et al., 2012*). However, a comprehensive understanding of how cold exposure and aging affect ASPC identity, adipogenesis, and adipocyte phenotypic switching remains elusive.

We applied ASPC lineage tracing, along with unbiased single-cell and single-nucleus RNA sequencing (scRNA-seq; snRNA-seq) to profile the beiging process and evaluate the impact of aging on this process. We found that aging modulates the gene program of fibroblastic ASPC populations and blocks the differentiation of these cells into beige adipocytes in vivo. snRNA-seq analysis revealed four types of adipocytes defined by different responses to cold exposure and aging: beige, *Npr3*-high, de novo lipogenesis (DNL)-low, and DNL-high. Notably, DNL-high adipocytes were defined by a marked induction of DNL genes during cold exposure in young compared to aged animals. A white adipocyte subpopulation in young mice was marked by expression of *Npr3*, which was also increased in adipocyte populations from aged mice. Altogether, this study shows that aging blocks cold-stimulated adipocyte reprogramming and ASPC adipogenesis, while implicating suppression of natriuretic peptide signaling and DNL as contributing to the aging-mediated decline in beige fat formation.

## Results
### Aging impairs iWAT beiging

To study the impact of aging on beige adipose tissue development, we exposed young (9-week-old) and middle aged (57-week-old) C57BL/6 mice to 6 °C for either 3 or 14 days. All mouse groups were first acclimated to 30 °C (thermoneutrality [TN]) for 3 weeks to reduce beige adipose tissue to baseline levels. Following acclimation, TN-housed mice remained at 30 °C; acute cold mice (3D) were transitioned to 6 °C after 11 days for the final 3 days; and chronic cold mice (14D) were moved to 6 °C for 2 weeks (*Figure 1A*). As expected, the aged mice weighed more and had larger iWAT depots than the young mice (*Figure 1—figure supplement 1A-B*). Cold exposure progressively increased the expression levels of thermogenic genes *Ucp1*, *Cidea*, *Dio2*, and *Ppargc1a* in iWAT from young mice, and the activation of these genes was significantly blunted in aged mice, especially at the 3D time point (*Figure 1B*). Immunofluorescence (IF) staining showed a robust induction of UCP1 protein in multilocular adipocytes of young iWAT at 3D of cold exposure, which was further increased at 14D. The induction of UCP1[+] beige adipocytes at 3D was severely reduced in aged animals, with few

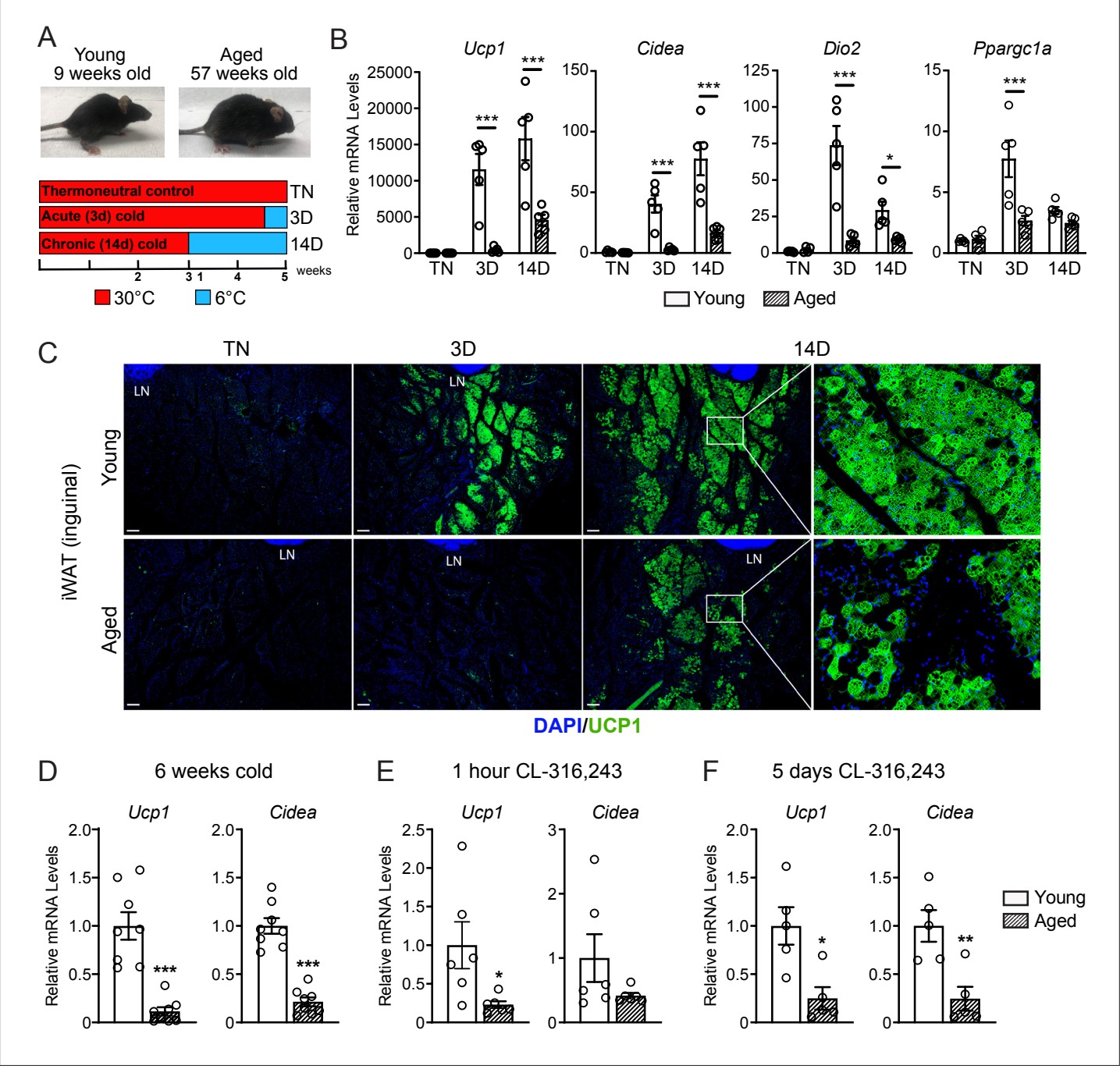

**Figure 1.** Aged mice exhibit decreased iWAT beiging in response to cold exposure or β3-agonist treatment. (**A**) Young (9-week-old) and aged (57-week-old) C57BL/6 mice were acclimated to 30 °C for 3 weeks, followed by two additional weeks either remaining at 30 °C (TN, thermoneutral), spending the last 3 days at 6 °C (3D, acute cold) or the last 14 days at 6 °C (14D, chronic cold). (**B**) Relative mRNA levels of thermogenic marker genes in mouse iWAT from (**A**), n=5. (**C**) Immunofluorescence analysis of UCP1 (green) and DAPI (blue) in iWAT sections from mice in (**A**), LN = lymph node. Scale bar 100 μm. (**D–F**) Relative mRNA levels of *Ucp1* and *Cidea* in iWAT from separate groups of young and aged mice that were either: exposed to 6 °C cold for 6 weeks (**D**), treated with CL-316,243 for 1 hr (**E**) or treated with CL 316,243 for 5 days (**F**). Data represent mean ± SEM, points represent biological replicates, two groups analyzed using a Student's t-test, and multiple conditions analyzed using a two-way ANOVA with a Tukey correction for multiple comparisons. Significance: not significant, p>0.05; * p<0.05 ** p<0.01; *** p<0.001.

The online version of this article includes the following figure supplement(s) for figure 1:

**Figure supplement 1.** Aging impairs WAT beiging.

UCP1 +adipocytes detected. At 14D, the beige adipocytes were morphologically similar in young and aged mice, although there were many fewer in aged animals (*Figure 1C*). At both ages, beige adipocytes were more prominent in the inguinal versus dorsolumbar region of iWAT, consistent with other reports (*Barreau et al., 2016*; *Chi et al., 2018*; *Dichamp et al., 2019*), and beiging was largely absent in the dorsolumbar region of aged mice (*Figure 1—figure supplement 1C–F*). To determine if the beiging response was delayed in aged mice, we exposed young and aged mice at 6 °C for 6 weeks. At this time point, the iWAT of aged mice exhibited a larger deficit in thermogenic gene expression compared to young animals (*Figure 1D*). Thermogenic gene levels in interscapular BAT were similar between young and aged mice at TN and after cold exposure, indicating that the inhibitory effects of aging were selective to WAT (*Figure 1—figure supplement 1E*).

Next, we examined beige fat formation in young and aged animals upon treatment with the β3-selective adrenergic agonist CL-316,243 (CL). CL acts in an adipose tissue autonomous manner to stimulate beige fat biogenesis, bypassing the central nervous system pathways that mediate the cold response. Acute CL treatment for only 1 hr increased Ucp1 expression in in iWAT of young mice to a much greater extent than in aged mice (*Figure 1E*). Chronic CL exposure for 5 days also induced much higher expression levels of Ucp1 and Cidea in iWAT of young compared to aged mice (*Figure 1F*). Taken together, these results demonstrate that beige adipose tissue induction is severely impaired in middle aged mice.

## Aging blocks beige adipogenesis from *Pdgfra*+ ASPCs

To determine the contribution of fibroblastic ASPCs to beige adipocytes during cold exposure, we performed lineage tracing using *Pdgfra-Cre^ERT2^*; *R26R^tdTomato^* reporter mice. *Pdgfra* expression marks multiple ASPC populations, including preadipocytes (*Merrick et al., 2019*; *Sakers et al., 2022*). Young and aged reporter mice were treated with tamoxifen for 5 days at TN (30 °C; 'pulse') to activate Cre and induce tdTomato expression in *Pdgfra*+ cells. Following a 9-day washout period, mice were transferred to 6 °C (cold) for 2 weeks ('chase'; *Figure 2A*). We observed near complete and specific labeling of ASPCs during the pulse period, with ~95% of PDGFRα+ cells in iWAT from young and aged mice displaying tdTomato expression (*Figure 2B*, *Figure 2—figure supplement 1*). The proportion of PDGFRα+ cells in iWAT was similar between young and aged mice (*Figure 2B*). No tdTomato-expressing adipocytes were observed after the pulse (*Figure 2—figure supplement 1*). After 14 days of cold exposure, we detected many newly developed beige adipocytes from ASPCs in young mice (visible as tdTomato+/UCP1+ multilocular adipocytes). By contrast, very few ASPC-derived (tdTomato+) adipocytes were detected in the beige fat areas of aged iWAT at day 14 (*Figure 2C*). Quantifying across the entire length of iWAT pads revealed that most beige adipogenesis occurred in the inguinal region and was ~12-fold lower in aged compared to young mice (*Figure 2D and E*). However, the overall contribution of *Pdgfra*+ ASPCs to beige adipocytes was relatively low, even in young animals, with <20% of beige adipocytes expressing tdTomato.

## Single-cell expression profiling of ASPCs

We previously identified three main fibroblastic ASPC populations in iWAT: DPP4+ cells, ICAM1+ preadipocytes, and CD142+ cells. All these cell types express *Pdgfra* and have the capacity to undergo adipogenic differentiation (*Merrick et al., 2019*). To test whether aging dysregulates one or more of these ASPC types, we performed scRNA-seq on stromal vascular cells from iWAT of young and aged animals, maintained at TN, or following transition to cold for 3 or 14 days (*Figure 1A*). ASPCs were enriched by removing immune (CD45+) cells using fluorescence activated cell sorting (FACS). We integrated the datasets from all conditions together and performed clustering analysis. The following cell populations were annotated based on their expression of cell-type-specific marker genes: four fibroblast populations (*Dpp4*+; *Icam1*+ preadipocytes; *Cd142*+, *Spp1*+), two populations of endothelial cells (*Pecam1*+); smooth muscle cells/pericytes (*Myh11*+, *Pdgfrb*+); Schwann cells (*Mpz*+); and residual immune cells (*Ptprc*+; *Figure 3A–C*). We did not identify any cell population specific to either aging or cold exposure. In this regard, we did not identify 'aging-dependent regulatory cells (ARCs)', which were previously defined as ASPCs expressing *Lgals3* and other inflammatory genes (*Figure 3—figure supplement 1*; *Nguyen et al., 2021*). The expression levels of identity markers of the ASPC populations were not modulated during cold exposure or aging (*Figure 3—figure supplement 1*).

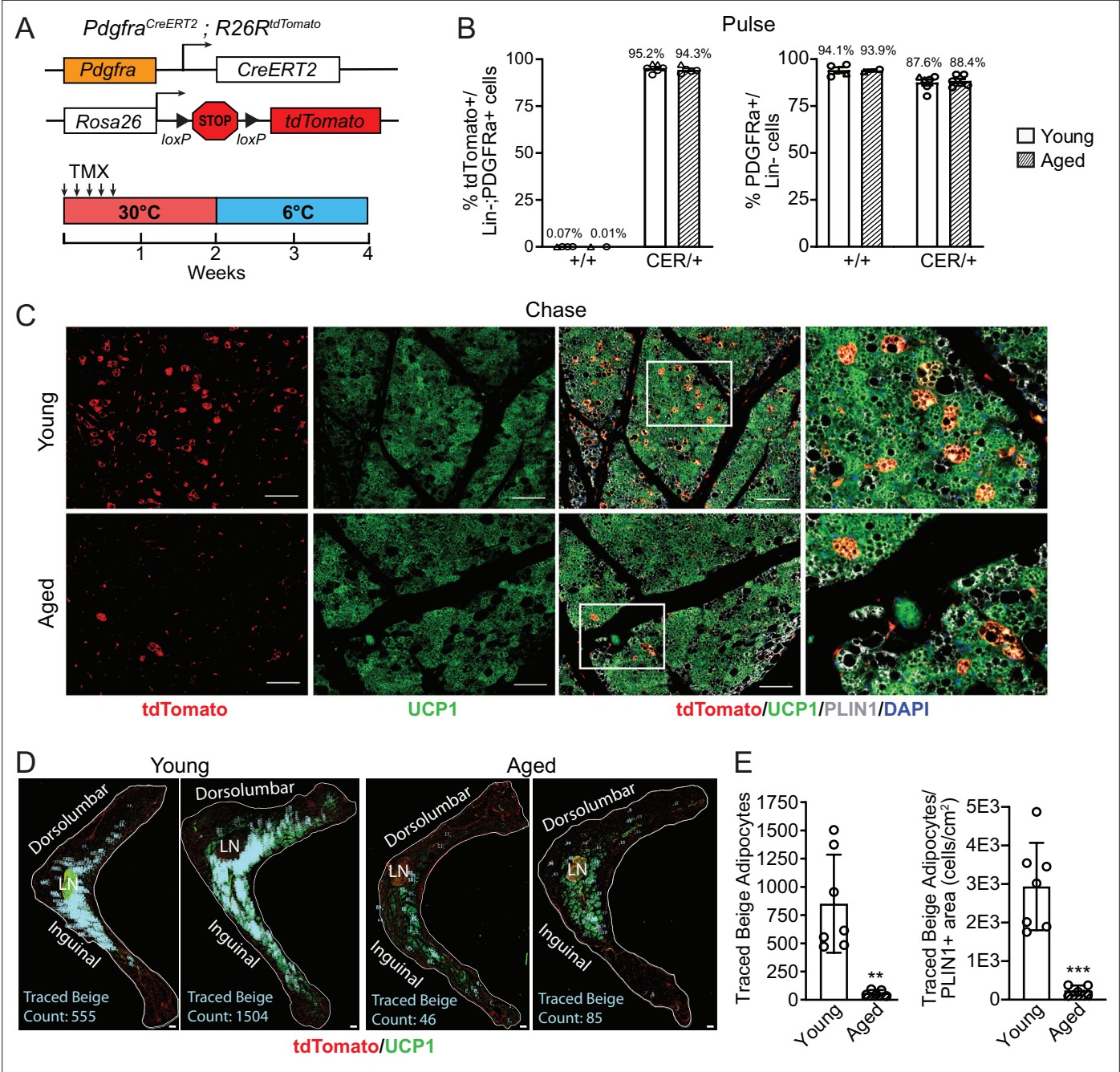

**Figure 2.** Aging blocks beige adipogenesis from fibroblastic ASPCs. (**A**) Schematic of *Pdgfra-CreERT2;R26R-tdTomato* reporter mouse model and lineage tracing paradigm. (**B**) Flow cytometry-based quantification showing proportions of tdTomato-expressing cells (as % of total Live, Lin- (CD45-/CD31-, PDGFRα+ cells)) (left) and PDGFRα+ cells (as % of total Live, Lin- cells) (right) in iWAT from young and aged Cre- (control, +/+), and Cre+ (CER) mice. n=6 young, 5 aged (Circles represent male mice, triangles represent female mice). (**C**) IF analysis of tdTomato (red), UCP1 (green), PLIN1 (white) and DAPI (blue) in iWAT from young and aged reporter mice after 14 days of 6 °C cold exposure (chase). Scale bar 100 μm. (**D**) Representative stitched images of full length iWAT histology slices from samples in (**C**) showing quantification of traced tdTomato+; UCP1 + multilocular (beige) adipocytes (blue numbers). LN = lymph node, scale bar 500 μm. (**E**) Quantification of traced beige adipocytes from (**D**) presented as total cell number (left) or proportion of PLIN1 + area (right), n=7 (young), n=5 (aged). Data represent mean ± SEM, points represent biological replicates, two groups analyzed using a Student's t-test, and multiple conditions analyzed with a two-way ANOVA with a Tukey correction for multiple comparisons. Significance: not significant, p>0.05; * p<0.05 ** p<0.01; *** p<0.001.

The online version of this article includes the following figure supplement(s) for figure 2:

**Figure supplement 1.** Aging blocks beige adipogenesis from PDGFRa +ASPCs.

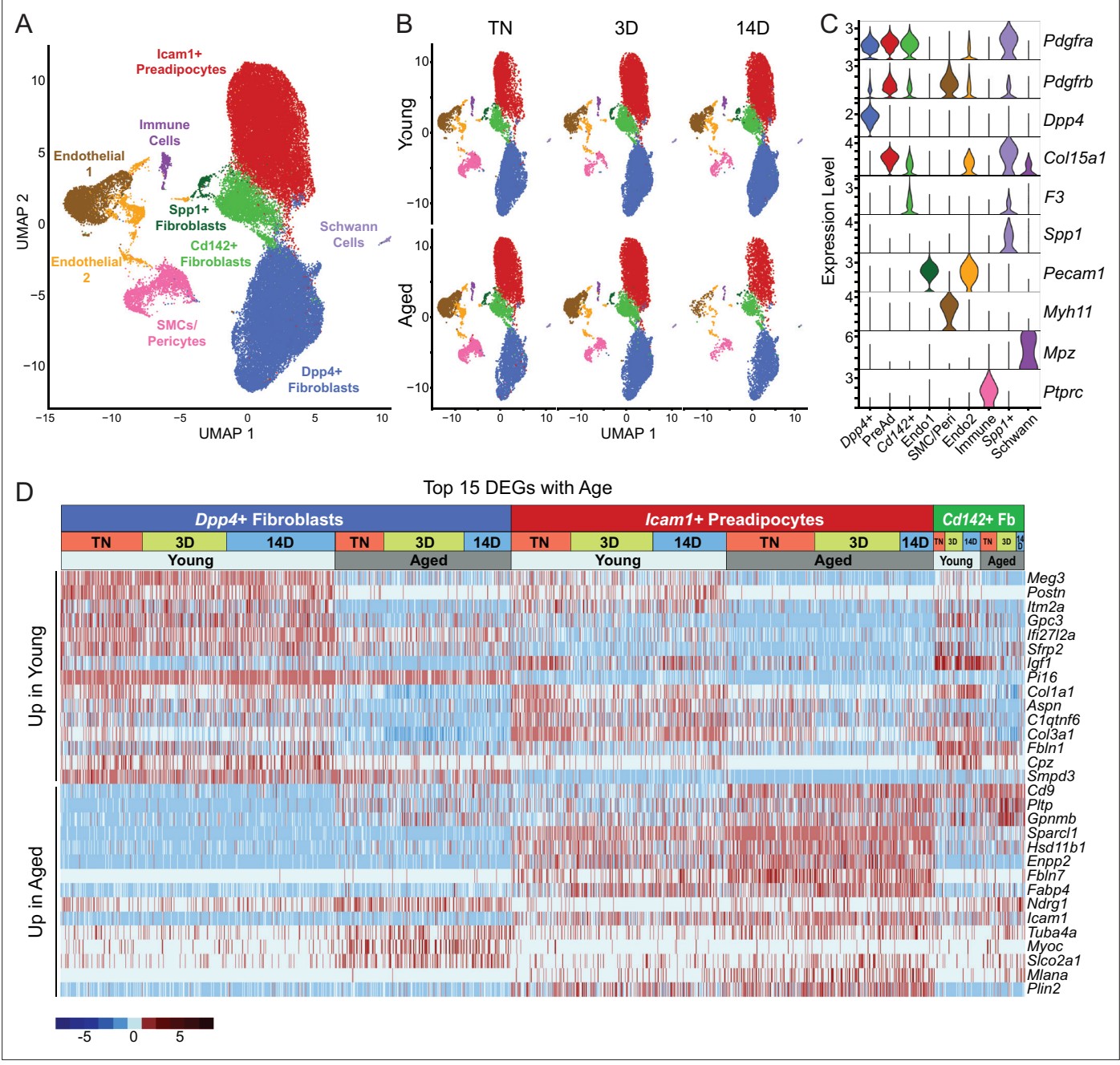

**Figure 3.** Single-cell expression profiling of ASPCs during iWAT beiging. (**A**) Integrated UMAP of gene expression in 54,987 stromal vascular cells (FACS depleted of CD45 +immune cells) from young and aged mouse groups detailed in *Figure 1A*. (**B**) UMAPs split by condition. (**C**) Violin plots showing the expression levels of representative marker genes for cell clusters. Y-axis=log-scale normalized read count. (**D**) Expression heatmap of the top differentially expressed genes in young vs. aged fibroblastic ASPCs (*Dpp4+*, *Icam1+* preadipocytes and *Cd142+* cells). Table shows expression of these genes in ASPC populations across temperature conditions (TN, cold 3D, cold 14D) from young and aged mice.

The online version of this article includes the following figure supplement(s) for figure 3:

**Figure supplement 1.** Single-cell expression profiling of ASPCs during iWAT beiging.

Differential gene expression analyses identified aging-modulated genes in ASPCs (*Figure 3D*). Notably, expression of *Cd9*, previously identified as a fibrogenic marker, was upregulated with age in *Dpp4⁺* cells and preadipocytes (*Marcelin et al., 2017*). *Pltp* and *Gpnmb* were also elevated by aging across all ASPC populations and temperature conditions. Genes downregulated by aging in all ASPC populations included *Meg3*, *Itm2a* and *Gpc3* and *Postn*. Of note, *Postn* encodes an extracellular

matrix protein that was previously reported to regulate adipose tissue expansion and decrease in expression during aging (*Graja et al., 2018*).

## ASPCs from aged mice are competent for beige adipogenesis ex vivo

We next evaluated if ASPCs from young and aged animals exhibit cell-autonomous differences in adipogenic differentiation capacity. We FACS-purified DPP4[+], ICAM1[+], and CD142[+] cells from the iWAT of young and aged mice, plated them in culture and induced adipocyte differentiation. Using a minimal differentiation stimulus consisting of insulin only (Min), ICAM1[+] and CD142[+] cells underwent more efficient differentiation into lipid droplet-containing adipocytes, and expressed higher levels of adipocyte genes (*Adipoq* and *Fabp4*) than DPP4[+] cells, consistent with prior work (*Figure 4A and B*; *Merrick et al., 2019*). DPP4[+] and CD142[+] cells from young and aged mice underwent adipocyte differentiation and induced adipocyte genes with equivalent efficiency. Unexpectedly, ICAM1[+] cells from aged mice exhibited greater differentiation capacity than those from young mice, as evidenced by higher expression levels of *Adipoq* and *Fabp4* (*Figure 4A and B*). Maximal stimulation with a full cocktail of adipogenic inducers (Max) produced similar and robust differentiation in all ASPC populations from young or aged mice (*Figure 4C and D*). To assess whether young and aged ASPCs behave differently when cultured as a mixed heterogeneous population, we isolated the stromal vascular fraction (SVF) for adipogenesis assays. Again, SVF cell cultures from young and aged mice displayed similar adipogenic differentiation efficiency following either Min or Max stimulation (*Figure 4E and F*). Finally, we treated differentiated adipocyte cultures with the pan-adrenergic agonist isoproterenol for 4 hours to evaluate thermogenic gene activation (i.e. beiging). Basal levels of *Ucp1* expression were lower in DPP4[+] cells compared to other ASPC types, but all ASPC populations activated *Ucp1* expression to similarly high levels in response to isoproterenol treatment and did not differ by age (*Figure 4G*). We also did not observe an aging-related difference in the levels of *Ucp1* induction in SVF-derived adipocyte cultures (*Figure 4H*). Together, these data suggest that the beige adipogenic capacity of ASPCs is not intrinsically compromised in aged mice, and therefore the in vivo deficit in beige adipogenesis could be due to non-ASPC-autonomous effects.

## Single-nucleus RNA sequencing uncovers adipocyte heterogeneity

To determine the effects of aging and cold exposure on adipocyte gene profiles, we performed snRNA-seq analyses of iWAT samples using the same experimental paradigm described above (*Figure 1A*). We integrated all the conditions together for analyses from two separate runs (*Figure 5A and C*). Similar cell types were captured as with scRNA-seq (*Figure 3A*), but with the addition of mature adipocyte populations. This dataset also has increased representation from immune cells since there was no negative selection against CD45[+] cells. As with the single-cell data set, we did not identify any aging-specific cell populations (*Figure 5—figure supplement 1*). However, we observed striking gene expression differences in the adipocyte cluster across age and temperature. Most obvious, and expectedly, was the emergence and expansion of a distinct beige adipocyte population, marked by expression of *Ucp1* and other thermogenic genes, during cold exposure (*Figure 5B*).

To focus on adipocyte responses, we reintegrated the snRNA-seq data using only the adipocytes, which revealed four main clusters (*Figure 5D–F*). All adipocyte clusters displayed similarly high mRNA levels of canonical adipocyte markers *Fabp4* and *Plin1*. Beige adipocytes, marked by high expression of thermogenic genes (i.e. *Ppargc1a*, *Esrrg*, *Cidea*, *Gk, Prdm16,* and *Ucp1*), were the most distinctive cluster and were largely absent at TN in young and aged mice. These cells began to appear in young mice after 3 days of cold exposure, and were further increased at 14 days. By contrast, in aged mice, beige cells were barely detectable at 3 days of cold exposure and were present at greatly reduced numbers than in young mice at 14 days (*Figure 5E*). This analysis also revealed three sub-populations of 'white' adipocytes. '*Npr3*-high' adipocytes were enriched for expression of *Npr3*, *Synpo2*, *Prr16*, and *Tshr*, expressed higher levels of white fat marker genes *Leptin* (*Lep*) and *Nnat*, and exhibited the lowest expression levels of thermogenic (beige) genes (*Gesta et al., 2007*; *Rosell et al., 2014*). Two additional white adipocyte clusters were designated as 'de novo lipogenesis (DNL)-low' and 'DNL-high' cells, both of which expressed lower levels of *Npr3* and shared selective expression of *Fgf14*. DNL-high cells uniquely expressed *Ces1f* and *Gsta3* and activated high levels of DNL pathway genes (i.e. *Fasn*, *Acss2* and *Acly*) upon cold exposure (*Figure 5F*). Interestingly, Adiponectin (*Adipoq*) was differentially expressed across adipocyte

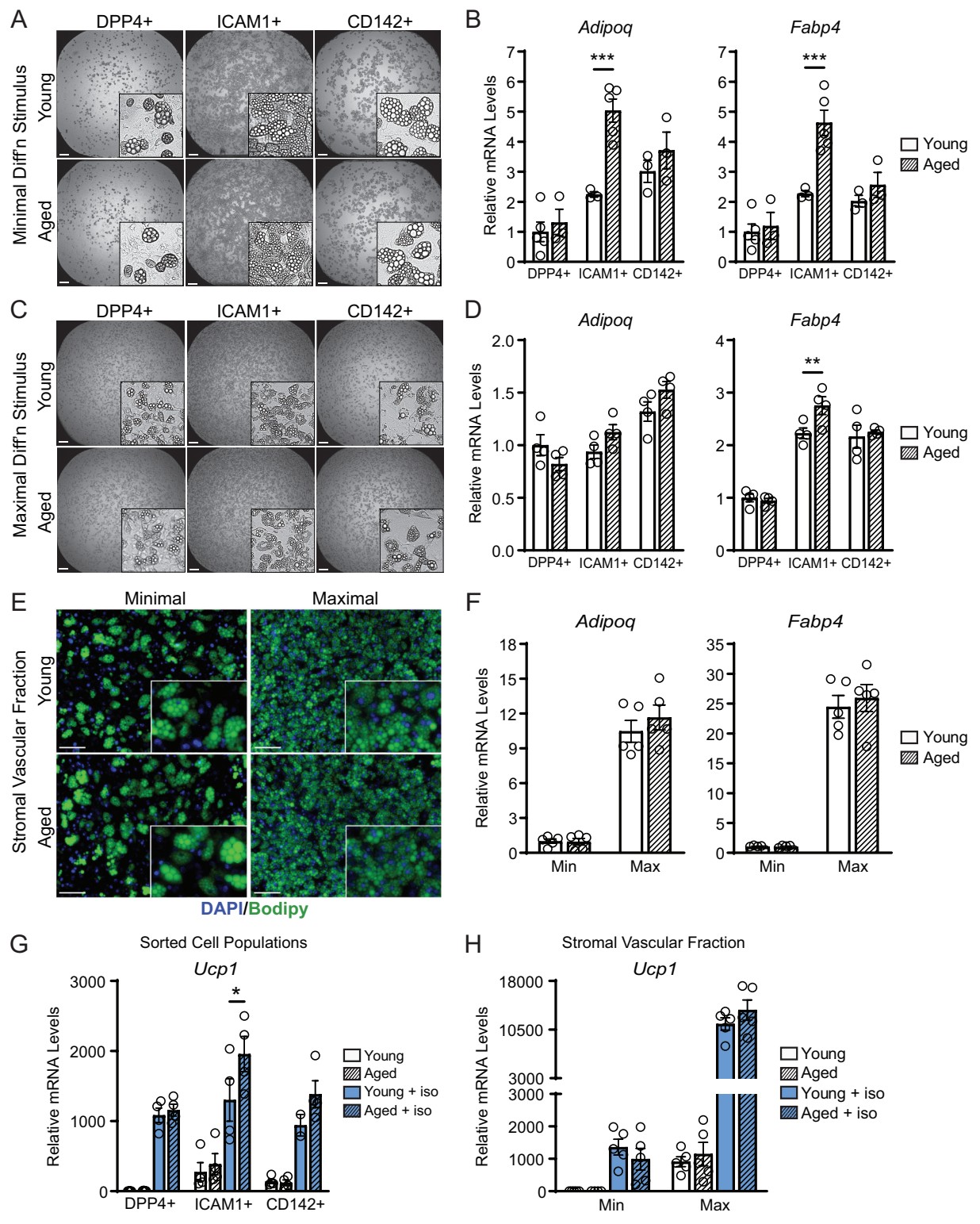

**Figure 4.** ASPCs from young and aged mice display similar beige adipogenic activity ex vivo. (**A, C**) Phase contrast images of DPP4+, ICAM1+ and CD142+ cells from iWAT of young and aged mice that were induced to undergo adipocyte differentiation with minimal (Min, **A**) or maximal (Max, **C**) induction cocktail for 8 days. Scale bar 200 μm. (**B, D**) mRNA levels of adipocyte marker genes *Adipoq* and *Fabp4* in cultures from (**A, C**). Data points represent separate wells, sorted from a pool of five mice (**A**) or sorted from two pools of two to three mice (**C**). (**E**) Stromal vascular fraction (SVF) cell cultures from the iWAT of young and aged mice were induced to differentiate for 8 days with Minimal or Maximal cocktail, followed by Bodipy (green) staining of lipid droplets and DAPI (blue) staining of nuclei. Scale bar 100 μm. (**F**) Relative mRNA levels of *Adipoq* and *Fabp4* in cultures from (**E**). Data

*Figure 4 continued on next page*

*Figure 4 continued*

points represent wells from individual mice, n=5. (**G, H**) Relative mRNA levels of *Ucp1* in adipocyte cultures from (**C, E**) with or without treatment with isoproterenol for 4 hr. Data points represent wells sorted from two pools of two to three mice (**G**) or wells from individual mice, n=5 (**H**). Data represent mean ± SEM, two groups analyzed using a Student's t-test, and multiple conditions analyzed with a two-way ANOVA with a Tukey correction for multiple comparisons. Significance: not significant, p>0.05; * p<0.05 ** p<0.01; *** p<0.001.

clusters, with higher levels in *Npr3*-high and DNL-high cells. Quantification of adipocyte nuclei from this data set suggested that the proportions of *Npr3*-high and DNL-high adipocytes remain stable across temperature, with aged mice having more *Npr3*-high adipocytes. The proportion of beige adipocytes increased during cold exposure, while DNL-low adipocytes decreased with cold exposure in both young and aged mice (***Figure 5G***).

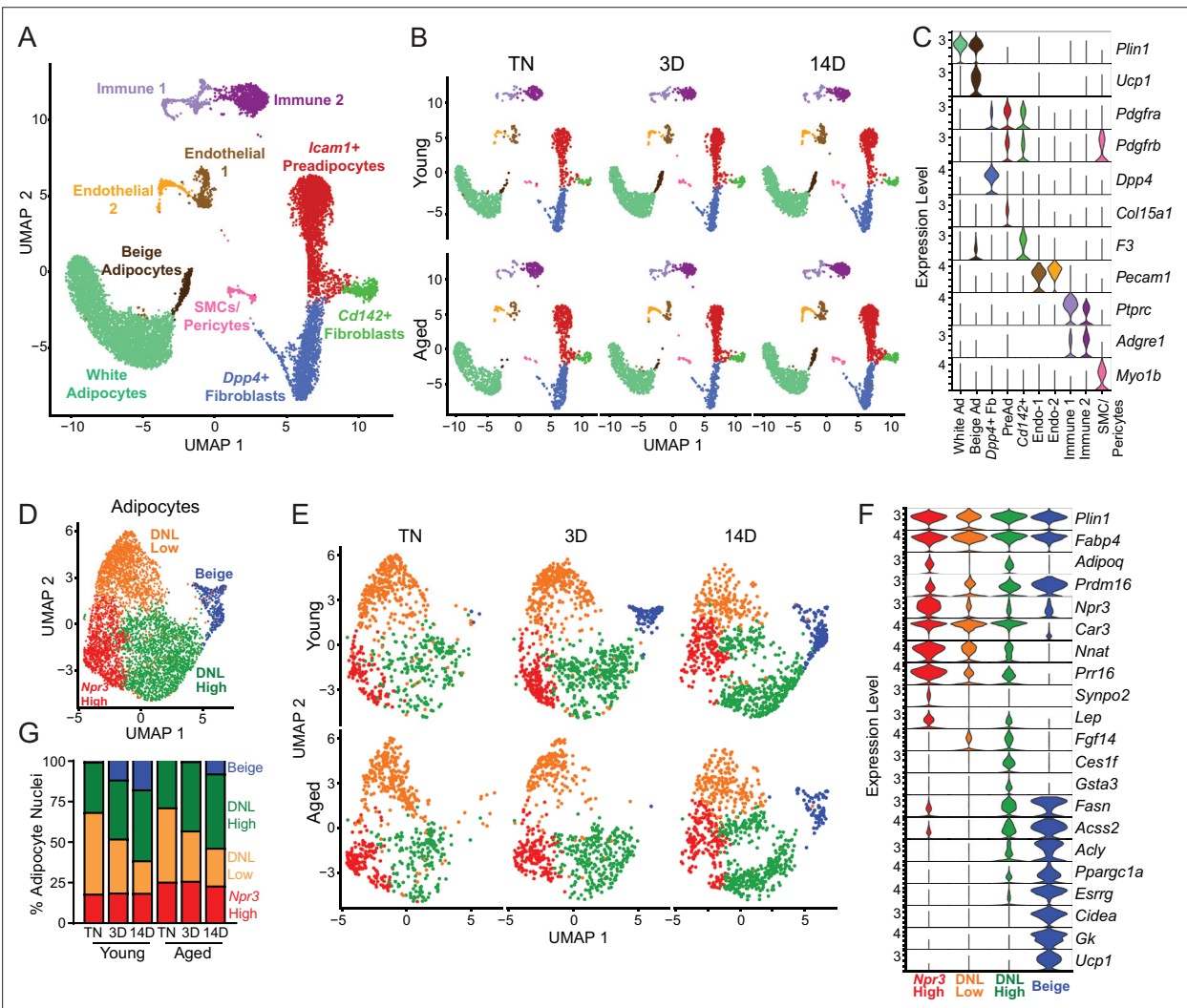

**Figure 5.** Single-nucleus expression profiling of adipocytes during the beiging process in young and aged mice. (**A**) Fully integrated UMAP of mRNA levels in 11,905 nuclei from iWAT of mouse groups detailed in ***Figure 1A***, n=2 mice per condition. (**B**) UMAPs split by condition. (**C**) Violin plots showing expression patterns of cell cluster-selective marker genes, Y-axis=log-scale normalized read count. (**D**) UMAP of gene expression in re-integrated adipocyte clusters including 4937 nuclei from (**A**) identifying four populations: *Npr3*-high, beige, DNL-low, and DNL-high. (**E**) Adipocyte UMAPs split by condition. (**F**) Violin plots showing expression patterns of selected genes in adipocyte populations, Y-axis=log-scale normalized read count. (**G**) Adipocyte nuclei numbers in each sample, plotted as percent of total adipocytes captured for that sample.

The online version of this article includes the following figure supplement(s) for figure 5:

**Figure supplement 1.** Single-nucleus expression profiling of iWAT during the beiging process.

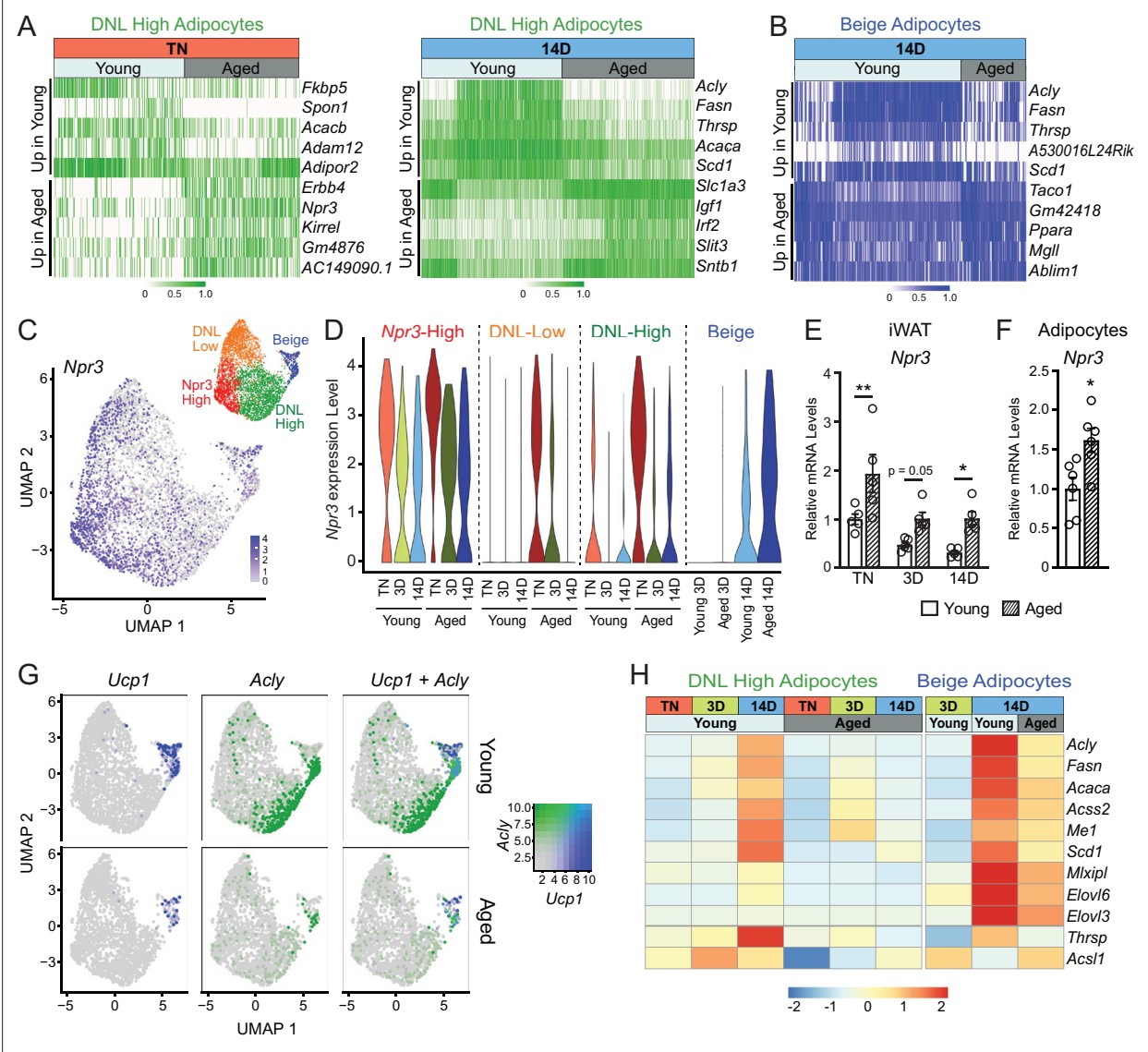

**Figure 6.** Aging blocks activation of the lipogenic gene program in adipocytes. (**A**) Expression heatmap of the top aging-regulated genes in DNL-high adipocytes at TN (left) and after 14 days of cold exposure (right). (**B**) Expression heatmap of the top aging-regulated genes in beige adipocytes after 14 days of cold exposure. (**C**) UMAP of *Npr3* mRNA levels in adipocyte populations (from *Figure 5D*). (**D**) Violin plots showing *Npr3* mRNA levels in adipocyte populations at TN (**T**), and at 3 and 14 days of cold exposure, Y-axis=log-scale normalized read count. (**E**) *Npr3* mRNA levels in iWAT from mouse groups described in *Figure 1A*, n=5. (**F**) *Npr3* mRNA levels in isolated adipocytes from TN- acclimated young and aged mice, n=6. (**G**) UMAPs of *Ucp1*, *Acly*, and their co-expression in adipocyte populations from young and aged mice. (**H**) Heatmap showing average expression of DNL genes in all nuclei from DNL-high and beige adipocytes per condition indicated in the top table. Data represent mean ± SEM, points represent biological replicates, two groups analyzed using a Student's t-test, and multiple conditions analyzed with a two-way ANOVA with a Tukey correction for multiple comparisons. Significance: not significant, p>0.05; * p<0.05 ** p<0.01; *** p<0.001.

The online version of this article includes the following figure supplement(s) for figure 6:

**Figure supplement 1.** Single-nucleus expression profiling of adipocytes during the beiging process.

## Aging dysregulates gene programming in adipocyte populations

To evaluate the global effects of cold exposure and aging on adipocytes, we performed differential gene expression analysis between young and aged adipocytes within each cluster. DNL-high and beige adipocytes exhibited the most dramatic expression changes between young and aged animals (*Figure 6A and B*, *Figure 6—figure supplement 1A and B*). At TN, DNL-high cells from aged animals expressed lower levels of several genes, including *Fkbp5*, *Spon1*, and *Adam12*. Interestingly, *Npr3*, in addition to marking *Npr3*-high cells, was increased by aging in DNL-high adipocytes and to a lesser

extent in other adipocyte populations (*Figure 6C and D*). In young animals, *Npr3* expression was downregulated by cold exposure in the three white adipocyte populations, and this downregulation was blunted in aged animals (*Figure 6D*). Gene expression analysis of whole iWAT pads confirmed that *Npr3* mRNA levels were progressively decreased by cold exposure and elevated in aged versus young mice under all temperature conditions (*Figure 6E*). *Npr3* expression levels were also increased in isolated primary adipocytes from aged relative to young mice (*Figure 6F*). Expression levels of the G-protein-coupled NP receptors *Npr1* or *Npr2* were not modulated by cold or aging in iWAT or iWAT adipocytes (*Figure 6—figure supplement 1*).

We also observed a striking activation of the DNL gene program (*Acly*, *Fasn*, *Acaca*, *Scd1*, etc.) in DNL-high and beige adipocytes during cold exposure (*Figure 6G and H*). The induction of these genes during cold exposure, exemplified by *Acly* expression, was a cluster-defining attribute of DNL-high cells, which did not express beige markers like *Ucp1* even after 14 days of cold exposure. Of note, we found two types of beige (*Ucp1*[+]) adipocytes, distinguished by the presence vs. absence of high DNL gene levels (i.e. *Ucp1*[+]; DNL[+] and *Ucp1*[+]; DNL[-]), with the latter arising first during cold exposure (3D vs. 14D) (*Figure 6G*, *Figure 6—figure supplement 1E*, F). Importantly, the induction of DNL genes was nearly completely blocked in DNL-high cells and reduced in beige cells of aged animals (*Figure 6G*). Indeed, the top aging downregulated genes in adipocytes from cold exposed mice correspond to DNL and related pathways, especially in DNL-high cells (*Figure 6—figure supplement 1G*). Lastly, at the whole tissue level, we observed robust induction of *Acly* in iWAT of young relative to aged mice with increasing duration of cold exposure (*Figure 6—figure supplement 1H*). Taken together, these results implicate the suppression of natriuretic peptide signaling and DNL in the aging-related impairment of beige fat formation.

## Discussion

Thermogenic adipose tissue activity declines during aging of mice and humans, correlating with increases in fat mass and susceptibility to cardiometabolic diseases (*Cypess et al., 2009*; *Saito et al., 2009*; *Pfannenberg et al., 2010*; *Yoneshiro et al., 2011*; *Rogers et al., 2012*; *Berry et al., 2017*; *Wang et al., 2019*; *Becher et al., 2021*). Our study provides a comprehensive unbiased profile of the adipose tissue beiging process and reveals pathways dysregulated by aging in ASPCs and adipocytes.

Beige adipocytes develop via the de novo differentiation of ASPCs or through activation of the thermogenic gene program in mature adipocytes. Previous studies defined three populations of fibroblastic ASPCs in iWAT, namely *Dpp4*[+] cells, *Icam1*[+] preadipocytes, and *Cd142*[+] cells. Aging or cold exposure did not induce dramatic shifts in either the proportions, or gene expression signatures of any of these ASPC types, suggesting that these cell populations are stably maintained across a range of conditions. In support of this, aging did not diminish the cell-intrinsic adipogenic capacities of these ASPC populations when subjected to adipogenesis assays ex vivo. Notably, we did not observe the emergence of aging-dependent regulatory cells (ARCs), previously described as modulated ASPCs co-expressing ASPC and immune marker genes, which have the capacity to suppress adipocyte differentiation (*Nguyen et al., 2021*). However, we did observe the induction of ARC-selective gene markers (i.e., *Lgals3*, *Cd36*) specifically in immune cells (*Ptprc*[+], *Adgre1*[+]) from aged mice in both our scRNA-seq and snRNA-seq datasets. This *Lgals3*/*Cd36* gene signature has also been described in Lin[+] macrophages and CD45[+] lipid-associated (LAM) macrophages (*Burl et al., 2018*; *Jaitin et al., 2019*). Overall, our results suggest that aging-induced alterations to the systemic milieu or adipose tissue environment are responsible for the block in beige adipogenesis.

Gene expression analyses identified several genes that were altered by aging across multiple ASPC types and temperature conditions. The top aging-upregulated gene was *Cd9*, which was previously identified as a marker of fibrogenic (fibrosis-generating) progenitor cells (*Marcelin et al., 2017*). *Cd9* encodes for a tetraspanin protein implicated in various processes that could affect adipogenesis, extracellular vesicle production, cell adhesion, inflammation, and platelet activation (*Brosseau et al., 2018*). Aging also upregulated the expression of *Pltp* and *Gpnmb*, which are both linked to the regulation of inflammation and fibrosis (*Prabata et al., 2021*; *Saade et al., 2021*). Conversely, *Meg3*, *Itm2a,* and *Postn* were consistently downregulated across all ASPC populations from aged versus young mice. Of note, Periostin (*Postn*) is an extracellular matrix protein that regulates adipose tissue lipid storage, and its levels were previously shown to decrease in several adipose tissue depots during aging (*Graja et al., 2018*).

We were surprised by the limited (<20%) contribution of fibroblastic (*Pdgfra*⁺) ASPCs, (which includes *Pparg*-expressing preadipocytes), to beige adipocytes during cold exposure. Of note, we also observed tdTomato⁺, unilocular white adipocytes upon cold exposure, suggesting the bi-potential fate of *Pdgfra*⁺ cells. Previous studies in mice using an adipocyte fate tracking system show that a high proportion of beige adipocytes arise via the de novo differentiation of ASPCs as early as 3 days of cold (*Wang et al., 2013*). However, the relative contribution from ASPC differentiation and direct adipocyte conversion to the formation of beige adipocytes depends highly on the experimental conditions, especially cold exposure history (*Shao et al., 2019*). Mice housed at TN from birth undergo high rates of de novo beige adipogenesis upon first cold exposure, whereas mice reared at room temperature acquire many 'dormant' beige adipocytes that can be re-activated by cold exposure (*Rosenwald et al., 2013*; *Shao et al., 2019*). Based on these findings, we presume that mature (dormant beige) adipocytes serve as the major source of beige adipocytes in our cold-exposure paradigm. However, long-term cold exposure also recruits smooth muscle cells to differentiate into beige adipocytes; a process that we did not investigate here (*Long et al., 2014*; *McDonald et al., 2015*; *Berry et al., 2016*; *Shamsi et al., 2021*).

The beiging process is associated with a dramatic remodeling of adipose tissue structure and metabolic function. We applied snRNA-seq analysis to investigate the cold response of iWAT adipocytes in young and aged animals, leading us to identify four adipocyte clusters: beige adipocytes and three 'white' subsets: *Npr3*-high, DNL-low and DNL-high adipocytes. *Npr3*-high adipocytes were enriched for expression of white fat-selective genes and exhibit the lowest levels of thermogenic genes (*Rosell et al., 2014*; *Ussar et al., 2014*). Interestingly, *Npr3* also upregulated by aging in all white adipocytes. Previous studies show that obesity also increases *Npr3* levels in adipose tissue of mice and humans (*Kovacova et al., 2016*; *Gentili et al., 2017*). NPR3 represses beige fat development and adipocyte thermogenesis by functioning as a clearance receptor for natriuretic peptides (NPs), thereby reducing their lipolytic and thermogenic effects (*Sengenès et al., 2000*; *Sengenes et al., 2003*; *Moro et al., 2004*; *Bordicchia et al., 2012*; *Coué et al., 2018*). Together, these results suggest that *Npr3*-high adipocytes may impede beige fat development in a cell non-autonomous manner by reducing NP signaling. Moreover, high NPR3 levels in aged animals could contribute to the block in beige fat development, and targeting this pathway may be a promising avenue to elevate beige fat activity.

We were also intrigued by the dramatic induction of DNL genes in beige adipocytes and DNL-high cells during cold exposure. DNL-high cells resemble an adipocyte subpopulation (mAd5), displaying enriched expression levels of *Acly* and *Acss2*, that was identified by *Emont et al., 2022*. Previous work established that cold stimulates opposing pathways of lipid oxidation and lipogenesis in thermogenic fat tissue (*Yu et al., 2002*; *Mottillo et al., 2014*; *Sanchez-Gurmaches et al., 2018*). The co-occurrence of these two processes is unusual and may provide a mechanism to ensure the continued availability of fatty acids to fuel thermogenesis and/or provide critical metabolic intermediates, such as acetyl-CoA. The Granneman lab demonstrated that high expression of the lipid catabolic enzyme MCAD and lipogenic enzyme FAS occurred in separate populations of iWAT adipocytes upon stimulation with a β3-adrenergic agonist for 3–7 days (*Lee et al., 2017*). We identified two subsets of UCP1⁺ beige adipocytes, distinguished by the presence vs. absence of high levels of DNL genes (i.e. *Ucp1*⁺; DNL-high and *Ucp1*⁺; DNL-low). Interestingly, the *Ucp1*⁺; DNL-high cells accumulated later during cold exposure (14D), suggesting that fully cold-adapted beige adipocytes express both pathways simultaneously. Of note, the induction of *Acly* and other lipogenic genes was very severely impaired in aged animals. Related to this point, Martinez Calejman and colleagues showed that *Acly* deficiency in brown adipocytes caused a whitened phenotype, coupled with an unexpected and unexplained reduction in *Ucp1* expression (*Martinez Calejman et al., 2020*). We speculate that high levels of ACLY may be required to support thermogenic gene transcription by supplying and efficiently shuttling acetyl-CoA for acetylation of histones or other proteins.

Aging is a complex process, and unsurprisingly, many pathways have been linked to the aging-related decline in beiging capacity. For example, increased adipose cell senescence, impaired mitochondrial function, elevated PDGF signaling and dysregulated immune cell activity during aging diminish beige fat formation (*Berry et al., 2017*; *Goldberg et al., 2021*; *Nguyen et al., 2021*; *Benvie et al., 2023*). Of note, older mice exhibit higher body and fat mass, which is associated with metabolic dysfunction and reduced beige fat development. While the effects of aging and altered body composition are difficult to separate, previous studies suggest that the beiging deficit in aged mice

is not solely attributable to changes in body weight (*Rogers et al., 2012*). Further studies, including additional time points across the aging continuum may help clarify the role of aging and ascertain when beiging capacity decreases.

In summary, this work shows that aging impairs beige adipogenesis through non-cell-autonomous effects on adipose tissue precursors and by disrupting adipocyte responses to environmental cold exposure. Expression profiling at the single-cell level reveals adipocyte heterogeneity, including two different types of UCP1[+] beige adipocytes. Finally, aging-dysregulated pathways, including natriuretic peptide signaling and lipogenesis, may provide promising targets for unlocking beige adipocyte development.

# Materials and methods

**Key resources table**

| Reagent type (species) or resource | Designation | Source or reference | Identifiers | Additional information |
|---|---|---|---|---|
| Genetic reagent (*M. musculus*) | C57BL/6 J | The Jackson Laboratory, Bar Harbor, ME | RRID:IMSR_ JAX:000664 | |
| Genetic reagent (*M. musculus*) | C57BL/6JN | NIA, Bethesda, MD | NA | |
| Genetic reagent (*M. musculus*) | *Rosa26 loxp-stop-loxp tdTomato* Reporter (Ai14) | The Jackson Laboratory, Bar Harbor, ME | RRID:IMSR_ JAX:007914 | |
| Genetic reagent (*M. musculus*) | *Pdgfra^CreERT2* | The Jackson Laboratory, Bar Harbor, ME | RRID:IMSR_ JAX:032770 | |
| Antibody | Rabbit polyclonal anti–red fluorescent protein (RFP) | Rockland, Pottstown, PA | 600-401-379, RRID:AB_2209751 | 1:500 |
| Antibody | Rabbit polyclonal anti-Perilipin (D418) | Cell Signaling, Denvers, MA | 3470, RRID:AB_2167268 | 1:200 |
| Antibody | Rabbit polyclonal anti- UCP1 | Specially made by AstraZeneca, Cambridge, UK | NA | 1:2000 |
| Antibody | Rabbit polyclonal Anti-mouse CD142 | Sino Biological, Chesterbrook, PA | R001 | 1:100 |
| Antibody | Goat polyclonal Anti-mouse CD142 | R & D Systems, Minneapolis, MN | AF3178, RRID:AB_2278143 | 1:50 |
| Antibody | Rat monoclonal Anti-mouse CD140a-(PDGFRα)-PECy7 | Biolegend, San Diego, CA | 135912, RRID:AB_2715974 | 1:100 |
| Antibody | Rat monoclonal Anti-mouse-CD31 (APC-Fire) | Biolegend, San Diego, CA | 102528, RRID:AB_2721491 | 1:1000 |
| Antibody | Rat monoclonal Anti-mouse CD45-allophycocyanin (APC/Cy7) | Biolegend, San Diego, CA | 103116, RRID:AB_312981 | 1:1000 |
| Antibody | Rat monoclonal Anti-mouse ICAM1-phycoerythrin (PE/Cy7) | Biolegend, San Diego, CA | 116122, RRID:AB_2715950 | 1:100 |
| Antibody | Rat monoclonal Anti-mouse CD26 (DPP-4)- fluorescein isothiocyanate (FITC) | Biolegend, San Diego, CA | 137806, RRID:AB_10663402 | 1:200 |
| Sequence-based reagent | mTbp | PMID:24703692 | NA | F-GAAGCTGCGGTACAATTCCAG R-CCCCTTGTACCCTTCACCAAT |
| Sequence-based reagent | mAdipoq | PMID:24703692 | NA | F-GCACTGGCAAGTTCTACTGCAA R-GTAGGTGAAGAGAACGGCCTTGT |

*Continued on next page*

*Continued*

| Reagent type (species) or resource | Designation | Source or reference | Identifiers | Additional information |
|---|---|---|---|---|
| Sequence-based reagent | mFabp4 | PMID:24703692 | NA | F-ACACCGAGATTTCCTTCAAACTG R-CCATCTAGGGTTATGATGCTCTTCA |
| Sequence-based reagent | mCidea | PMID:24703692 | NA | F-TGCTCTTCTGTATCGCCCAGT R-GCCGTGTTAAGGAATCTGCTG |
| Sequence-based reagent | mPgc1a | PMID:24703692 | NA | F-CCCTGCCATTGTTAAGACC R-TGCTGCTGTTCCTGTTTTC |
| Sequence-based reagent | mUcp1 | PMID:24703692 | NA | F-ACTGCCACACCTCCAGTCATT R-CTTTGCCTCACTCAGGATTGG |
| Sequence-based reagent | mDio2 | PMID:24703692 | NA | F-CAGTGTGGTGCACGTCTCCAATC R-TGAACCAA AGTTGACCACCAG |
| Sequence-based reagent | mAcly | PMID:31141698 | NA | F-GAGTGCTATTGCGCTTCCC R-GGTTGCCGAAGTCACAGGT |
| Sequence-based reagent | mNpr3 | This Paper | NA | F-TTTTCAGGAGGAGGGGTTGC R-ACACATGATCACCACTCGCT |
| Sequence-based reagent | mNpr1 | MGH PrimerBank | Primer Bank ID: 113930717 c1 | F-GCTTGTGCTCTATGCAGATCG R-CCTCGACGAACTCCTGGTG |
| Sequence-based reagent | mNpr2 | MGH PrimerBank | Primer Bank ID: 118129825 c2 | F-CATGACCCCGACCTTCTGTTG R-CGAACCAGGGTACGATAATGCT |
| Commercial assay or kit | ABI High-Capacity cDNA Synthesis kit | Applied Biosystems, Waltham, MA | 4368813 | |
| Commercial assay or kit | Purelink RNA Mini columns | Invitrogen, Waltham, MA | LT-12183018 | |
| Commercial assay or kit | TSA TMR Tyramide Reagent Pack | Akoya Biosciences, Marlborough, MA | NEL742001KT | |
| Commercial assay or kit | TSA Fluorescein Tyramide Reagent Pack | Akoya Biosciences, Marlborough, MA | NEL741001KT | |
| Commercial assay or kit | Bulls Eye Decloaking Buffer | Biocare, Pacheco, CA | BULL1000 MX | |
| Commercial assay or kit | AbC Total Antibody Compensation Bead Kit | BioLegend,San Diego, CA | A10497 | |
| Commercial assay or kit | Biotium Mix-n-Stain CF647 | Sigma, Burlington, MA | MX647S100 | |
| Commercial assay or kit | PicoPure RNA Isolation Kit | Invitrogen, Waltham, MA | KIT0204 | |
| Commercial assay or kit | Qubit dsDNA High Sensitivity assay kit | ThermoFisher, Waltham, MA | Q32851 | |
| Commercial assay or kit | DNA High Sensitivity Bioanalyzer Chip (Agilent) | Agilent, Santa Clara, CA | 5067–4626 | |
| Software, algorithm | Graphpad Prism | Graphpad, San Diego, CA | RRID:SCR_002798 | |
| Software, algorithm | Adobe Illustrator | Adobe, San Jose, CA | RRID:SCR_010279 | |
| Software, algorithm | Adobe Photoshop | Adobe, San Jose, CA | RRID:SCR_014199 | |
| Software, algorithm | Image J | PMID:22743772 | RRID:SCR_003070 | |
| Software, algorithm | Cell Ranger | 10 x Genomics | RRID:SCR_017344 | |
| Software, algorithm | Seurat | PMID:34062119 | RRID:SCR_016341 | |
| Software, algorithm | bcl2fastq | Illumina | RRID:SCR_015058 | |

*Continued on next page*

*Continued*

| Reagent type (species) or resource | Designation | Source or reference | Identifiers | Additional information |
|---|---|---|---|---|
| Software, algorithm | Cumulus | PMID:32719530 | RRID:SCR_021644 | |
| Software, algorithm | FACSDiva Softward | Becton Dickinson, Franklin Lakes, NJ | RRID:SCR_001456 | |
| Other | Tamoxifen (Free Base) | Sigma, Burlington, MA | T5648 | Synthetic estrogen receptor antagonist used to activate Cre. |
| Other | Corn Oil | Sigma, Burlington, MA | C8267 | Vehicle solution for tamoxifen. |
| Other | 16% Paraformaldehyde | EMS, Hatfield, PA | 15710 | Fixative used for tissue histology |
| Other | TRIzol | Invitrogen, Waltham, MA | 15596018 | Phenol-based solution used for nucleic acid extraction |
| Other | CL-316,243 | Sigma, Burlington, MA | C5976 | Agonist of Beta3-adrenergic receptor |
| Other | 4',6-Diamidine-2'-phenylindole dihydrochloride (DAPI), 1:10,000 | Roche, Basel, Switzerland | 10236276001 | Fluorescent stain for DNA/nuclei |
| Other | Bovine Serum Albumin, fraction V, fatty-acid free | Gold Biotechnology, St. Louis, MO | A-421–250 | Protein carrier for small molecules |
| Other | DMEM/F12 | Fisher Scientific, Waltham, MA | 11320033 | Basal cell culture medium |
| Other | Fetal Bovine Serum | Omega Scientific, Tarzana, CA | FB-11, Lot 401714 | For cell culture |
| Other | Primocin | InvivoGen, San Diego, CA | ant-pm-2 | Anti-microbial for cell culture |
| Other | PCR Master Mix, Power SYBR Green | Applied Biosystems, Waltham, MA | 4367659 | Kit for qRT-PCR |
| Other | HBSS, 1 X | Fisher Scientific, Waltham, MA | 14175079 | Hank's Balanced Salt Solution |
| Other | Dispase II | Roche, Basel, Switzerland | 4942078001 | Enzyme used for adipose tissue digestion |
| Other | Collagenase, Type 1 | Worthington, Lakewood, NJ | LS004197 | Enzyme used for adipose tissue digestion |
| Other | Red Blood Cell Lysis Buffer, 10 x | BioLegend, San Diego, CA | 420302 | For lysing red blood cells during cell isolations |
| Other | Human Insulin, Novolin | Novo Nordisk, Bagsvaerd, Denmark | 183311 | Used for cell culture studies |
| Other | Dexamethasone | Sigma-Aldrich, Burlington, VT | D4902 | Glucorticoid Receptor agonist |
| Other | 3-isobutyl-1-methylxanthine (IBMX) | Sigma-Aldrich, Burlington, VT | I7018 | Chemical used to Increase cAMP levels, used in adipocyte differentiation cocktail |
| Other | Rosiglitazone | Cayman Chemical, Ann Arbor, MI | 11884 | Synthetic PPARgamma activator |
| Other | Indomethacin | Sigma-Aldrich, Burlington, VT | I8280 | Chemical used in adipocyte differentiation cocktail |
| Other | 3,30,5-Triiodo-L-thyronine sodium salt (T3) | Sigma-Aldrich, Burlington, VT | T6397 | Thyroid Receptor agonist |
| Other | isoproterenol | Sigma-Aldrich, Burlington, VT | I6504 | Pan beta-adrenergic receptor agonist |
| Other | Bodipy 493/503 | Invitrogen, Waltham, MA | D3922 | Fluorescent dye for neutral lipids |

*Continued on next page*

*Continued*

| Reagent type (species) or resource | Designation | Source or reference | Identifiers | Additional information |
|---|---|---|---|---|
| Other | Hoechst 33342 | Thermo Fisher, Waltham, MA | 62249 | DNA stain |
| Other | Protector RNase Inhibitor | Roche, Basel, Switzerland | 3335399001 | Used for RT-PCR |

## Mice

All animal procedures were approved and performed under the guidance of the University of Pennsylvania Institutional Animal Care and Use Committee (IACUC) (protocol #805649). Young (4 weeks) and aged (52 weeks) C57BL/6 male mice were obtained from the National Institute of Aging (C57BL/6JN) or Jackson Laboratories (C57BL/6 J, stock number 000664). Mice were housed at 30 °C for 3 weeks, then were either: maintained at 30 °C for 2 weeks (TN); kept at 30 °C for 11 more days before moving to 6 °C for 3 days (3D cold) or moved to 6 °C for 14 days (14D cold). Mice were single housed during the final 2-week temperature treatment and provided with a nestlet and shepherd shack. For experiments with CL316,243 (CL, Sigma-C5976), mice were housed at 30 °C for 5 weeks, followed by intraperitoneal (IP) injection of 1 mg/kg/d CL either 1 hr prior to tissue harvest or for 5 days. $Pdgfra^{CreERT2}$ mice were obtained from Dr. Brigid Hogan (Duke University) (*Chung et al., 2018*) and crossed with $Rosa26^{tdTomato}$ (strain: B6.Cg-Gt(ROSA)26Sortm14(CAG-tdTomato)Hze/J, stock no. 007914). To induce Cre activity, tamoxifen (Sigma, T5648) dissolved in corn oil (Sigma, C8267) was injected intraperitonially (IP) into mice at a dose of 100 mg/kg/d for 5 days. For all iWAT processing other than histology, the inguinal lymph node was removed.

## Histology and immunofluorescence

Tissues were fixed overnight in 4% paraformaldehyde, washed with PBS, dehydrated in ethanol, paraffin-embedded and sectioned. Following deparaffinization, slides were subjected to heat antigen retrieval in a pressure cooker with Bulls Eye Decloaking buffer (Biocare), unless otherwise noted. Slides were incubated in primary antibody overnight and secondary antibody conjugated to peroxidase and then developed using Tyramide Signal Amplification (TSA, Akoya Biosciences). Samples were stained with either hematoxylin and eosin or the following antibodies: anti-red fluorescent protein (RFP) (rabbit; 1:500; Rockland #600-401-379), anti-UCP1 (rabbit, 1:2000, AstraZeneca), and anti-PLIN1 (rabbit, 1:200 Cell Signaling #3470). Slides were imaged on an inverted fluorescence microscope (Keyence BZ-X710). For quantification of tdTomato-expressing adipocytes, full-length iWAT slices were tile imaged, stitched, exported as a BigTiff, and quantified in a blinded-manner using the Count Tool in Photoshop (Adobe).

## Isolation of stromal vascular cells (SCVs) and adipocytes

### SVCs

As previously described (*Merrick et al., 2019*; *Wang et al., 2019*), iWAT tissue was dissected, minced gently and digested with Collagenase Type I (1.5 units/ml; Worthington) and Dispase II (2.4 units/ml; Roche) in DMEM/F12 containing 1% fatty acid-free bovine serum albumin (Gold Biotechnology) in a gentleMACS dissociator (Miltenyi Biotec) on program '37 MR ATDK-1'. The digestion was quenched with DMEM/F12 containing 10% FBS, and the dissociated cells were passed through a 100 µm filter and spun at 400 x $g$ for 4 min. The pellet was resuspended in red blood cell lysis buffer (BioLegend), incubated for 4 min at RT, then quenched with DMEM/F12 containing 10% serum. Cells were passed through a 70 µm filter, spun, resuspended, then passed through a final 40 µm filter, spun at 400 x $g$ for 4 min and plated or underwent further processing for FACS. Mice were not pooled unless indicated.

### Adipocytes

Tissue went through the same process as above, except after digestion and quenching, adipocyte/SVF slurry was filtered through a 200 µm filter and centrifuged at 50 x $g$ for 3 min at RT. Using a 20 mL syringe and 1.5-inch, 25 G needle, media containing the SVCs was removed from below the adipocytes (and saved if concurrently isolating SVCs), leaving only the adipocytes in the tube. Adipocytes

were washed twice with the same media as quenching, transferred to 2 mL tubes, spun a final time, media was removed from below the adipocytes again, and TRIzol was added for RNA extraction. Mice were not pooled.

## FACS

DPP4$^+$, ICAM1$^+$, and CD142$^+$ cells were isolated as previously described (*Merrick et al., 2019*). Briefly, SVCs from the subcutaneous adipose of mice (n=2–5) were pooled and resuspended in FACS buffer (HBSS containing 3% FBS; Fisher), then incubated for 1 hr at 4 °C with the following antibodies: CD26 (DPP4)-fluorescein isothiocyanate (FITC) (Biolegend, 137806; 1:200), anti-mouse ICAM1-phycoerythrin (PE)/Cy7 (Biolegend, 116122; 1:100), anti-mouse CD45-allophycocyanin (APC)/ Cy7 (Biolegend, 103116; 1:1000), anti-mouse CD31-APC-Fire (Biolegend, 102528; 1:1000), and anti-mouse CD142 (Sino Biological, 50413-R001, 1:100; or R&D Systems, AF3178, 1:50). Anti-mouse CD142 antibodies were conjugated with Biotium Mix-n-Stain CF647 (Sigma, MX647S100). For lineage tracing pulse analysis, SVCs were isolated from individual mice without pooling. SVCs were stained with anti-mouse CD31, anti-mouse CD45, and anti-mouse CD140a (PDGFRΑ) (PE/Cy7) (Biolegend, 135912; 1:100). In all FACS experiments, cells were stained with 4',6-diamidino-2-phenylindole (DAPI) (Roche, 10236276001; 1:10,000) for 5 min, then washed three times with FACS buffer to remove unbound antibodies. Cells were sorted with a BD FACS Aria cell sorter (BD Biosciences) equipped with a 100 µm nozzle and the following lasers and filters: DAPI, 405 and 450/50 nm; FITC, 488 and 515/20 nm; mTomato, 532 and 610/20 nm; PE/Cy7, 532 and 780/60 nm; CF647, 640 and 660/20 nm; and APC/Cy7 and APC-Fire, 640 and 780/60 nm. All compensation was performed at the time of acquisition in Diva software by using compensation beads (BioLegend, A10497) for single-color staining and SVCs for negative staining and fluorescence (DAPI and tdTomato).

## Cell culture and differentiation
### Adipocyte precursor cells
All cells were cultured in DMEM/F12 containing 10% FBS and Primocin (50 ng/ml; InvivoGen, ant-pm-1). DPP4$^+$, ICAM1$^+$, and CD142$^+$ populations were FACS purified, plated on CellBind 384-well plates (Corning) at 15–25 K cells/well, and incubated for 48 (25K cells) to 72 hr (15 K cells) to facilitate attachment before the induction of adipogenic differentiation. For whole SVF, SVCs were isolated and plated in a 48 well CellBind plate (Corning) at a high confluency of one mouse per 18 wells. No cells were passaged after plating to maintain adipogenic competency. Differentiation was carried out with either maximum adipogenic cocktail, max: 500 µM isobutylmethylxanthine (Sigma, I7018), 10 µM dexamethasone (Sigma, D4902), 125 µM indomethacin (Sigma, I8280), 1 µM rosiglitazone (Cayman Chemical, 11884), 1 nM T3 (Sigma, T6397), and 20 nM insulin (Novolin) or a minimal adipogenic cocktail, min: 20 nM insulin. For the max adipogenic cocktail induction, cells were incubated with cocktail for 2 days and then transferred to adipogenic maintenance medium for the remaining 6 days (1 µM rosiglitazone, 1 nM T3, and 20 nM insulin). For all conditions, medium was changed every 2 days, and cells were harvested on day 8 of differentiation. For drug treatments, cells were treated for 4 hr on day 8 with 1 µM isoproterenol (Sigma, I6504). Adipogenesis was assessed by staining with Biodipy 493/503 (Invitrogen, D3922) for lipid droplet accumulation and Hoechst 33342 (Thermo Fisher, 62249) for nuclei number. The cells were imaged on a Keyence inverted fluorescence microscope (BZ-X710) by using DAPI (excitation, 360/40 nm; emission, 460/50 nm) and green fluorescent protein (excitation, 470/40 nm; emission, 525/50 nm) filters. Individual wells were imaged in their entirety at ×4 magnification, and at 20 x to see morphology. 384-well plates were not stained and imaged in brightfield due to low cell number recovery from FACS prior to RNA extraction.

## RNA Extraction, qRT-PCR and RNA Sequencing
### RNA Extraction
Total RNA was extracted using TRIzol (Invitrogen) combined with PureLink RNA Mini columns (Thermo Fisher, 12183025) for tissue and SVC cells or by PicoPure RNA Isolation Kit (Applied Biosystems, KIT0204) for 384-well plate populations and adipocytes. Prior to the addition of chloroform, all tissue and primary adipocytes in TRIzol included an extra spin at max speed for 10 min at RT, then TRIzol was removed from below the lipid layer to avoid lipid contamination disrupting the subsequent phase separation with chloroform. Chloroform was added to the lipid-free TRIzol, spun for 15 min at

12,000 x *g* and the aqueous layer was removed and added to columns. mRNA was quantified using a Nanodrop and reverse transcribed to cDNA using the ABI High-Capacity cDNA Synthesis kit (ABI, 4368813). Real-time PCR was performed on a QuantStudio5 qPCR machine using SYBR green fluorescent dye (Applied Biosystems). Fold changes were calculated using the ddCT method, with TATA binding Protein (*Tbp*) mRNA serving as a normalization control.

## Single-cell RNA-seq Samples

Cells were flow sorted to isolate live (DAPI⁻) cells and remove debris. We enriched non-immune cells by sorting out CD45⁺ cells. Next-generation sequencing libraries were prepared using the Chromium Next GEM Single Cell 3' Reagent kit v3.1 (10x Genomics, 1000121) per manufacturer's instructions. Libraries were uniquely indexed using the Chromium Single Index Kit T Set A, pooled, and sequenced on an Illumina NovaSeq 6000 sequencer in a paired-end, dual indexing run by the CHOP Center for Applied Genomics at the University of Pennsylvania. Sequencing for each library targeted 20,000 mean reads per cell.

## Single nucleus RNA-seq samples

Nuclei were isolated from frozen mouse iWAT samples as previously described, with the following modifications to integrate hash multiplexing and FANS-assisted nuclear quality thresholding and sample pooling (*Drokhlyansky et al., 2020*; *Slyper et al., 2020*). Briefly, 300 mg of flash-frozen adipose samples were held on dry ice until immediately before nuclei isolation, and all sample handling steps were performed on ice. Each sample was placed into a gentleMACS C tube (Miltenyi Biotec, 130-093-237) with 2 mL freshly prepared TST buffer (0.03% Tween 20 (Bio-Rad), 0.01% Molecular Grade BSA (New England Biolabs), 146 mM NaCl (Thermo Fisher Scientific), 1 mM CaCl₂ (VWR International), 21 mM MgCl₂ (Sigma Aldrich), and 10 mM Tris-HCl pH 7.5 (Thermo Fisher Scientific) in ultrapure water (Thermo Fisher Scientific)) with 0.2 U/µL of Protector RNase Inhibitor (Sigma-Aldrich, RNAINH-RO). gentleMACS C tubes were then placed on the gentleMACS Dissociator (Miltenyi Biotec) and tissue was dissociated by running the program 'mr_adipose_01' three times, and then incubated on ice for 10 min. Lysate was passed through a 40 µm nylon filter (CellTreat) and collected into a 50 mL conical tube (Corning). Filter was rinsed with 3 mL of freshly prepared ST buffer (146 mM NaCl, 1 mM CaCl₂, 21 mM MgCl₂; 10 mM Tris-HCl pH 7.5) with 0.2 U/µL RNase Inhibitor, and collected into the same tube. Flow-through was passed through a 20 µm pre-separation filter (Miltenyi Biotec) set on top of a 5 mL FACS tube (Corning) and collected into the same tube. Suspension was centrifuged in a swinging-bucket centrifuge (Eppendorf) at 500 × *g* for 5 min at 4 °C with brake set to low. Following centrifugation, supernatant was removed and 5 mL of PBS pH 7.4 (Thermo Fisher Scientific) with 0.02% BSA and 0.2 U/µL RNase Inhibitor was added without resuspending the nuclear pellet. Sample was centrifuged again at 500 × *g* for 5 minutes at 4 °C with brake set to low. Following centrifugation, supernatant was removed, and the nuclear pellet was resuspended in 1 mL PBS-0.02% BSA with 0.2 U/µL RNase Inhibitor. Each sample was split into two 500 µL aliquots and transferred to new 5 mL FACS tubes for subsequent hashing. Each aliquot of resuspended nuclei was stained with NucBlue (ThermoFisher, R37605), labeled with 1 µg of a unique TotalSeq anti-Nuclear Pore Complex Proteins Hashtag Antibody (Biolegend), and then incubated on ice for 30 min. Suspension was centrifuged at 500 × *g* for 5 min at 4 °C with brake set to low. Following centrifugation, 450 µL of supernatant was removed and the nuclear pellet was resuspended in 450 µL PBS-0.02% BSA with 0.2 U/µL RNase Inhibitor. For nuclear quality thresholding, fluorescence-activated nuclear sorting (FANS) was implemented to collect 4,000–4,300 nuclei from hashtagged aliquots directly into a shared well of a 96-well PCR plate (Thermo Scientific) containing 24.6 µL of 10 X RT Reagent B with 1 U/uL RNase Inhibitor on a Beckman Coulter MoFlo AstriosEQ fitted with a 70 µm nozzle. High-quality nuclei were selected by initial gating at 360 nm with laser filter 405-448/59 followed by SSC-H and FSC-H to remove doublets and unlysed cells. Once all sample aliquots were FANS-sorted, the pool of 43,000 nuclei was loaded on the 10 x Chromium controller (10 x Genomics) according to the manufacturer's protocol. cDNA and gene expression libraries were generated according to the manufacturer's instructions (10 x Genomics). Libraries of hashtag oligo fractions were generated according to the manufacturer's instructions (Biolegend). cDNA and gene expression library fragment sizes were assessed with a DNA High Sensitivity Bioanalyzer Chip (Agilent). cDNA and gene expression libraries were quantified using the Qubit dsDNA High Sensitivity assay kit (Thermo Fisher, Q32854). Gene expression libraries were

multiplexed and sequenced on the Nextseq 500 (Illumina) using a 75-cycle kit and the following read structure: Read 1: 28 cycles, Read 2: 55 cycles, Index Read 1: 8 cycles.

## Bioinformatics analysis

### Single-cell RNA sequencing

Data was processed using the Cell Ranger pipeline (10 x Genomics, v.3.1.0) for demultiplexing and alignment of sequencing reads to the mm10 transcriptome and creation of feature-barcode matrices. The cell ranger output files were read into R (version 4.1.1) and processed utilizing the standard Seurat CCA integrated workflow (version 4.3.0). Each of the six samples went through a first phase of filtering, where only cells that recorded more than 200 features and only features present in a minimum of 3 cells were kept. Each sample was filtered prior to downstream analysis on nCount_RNA, nFeature_RNA, and mitochondrial percentages. Samples were then normalized using a LogNormalization method with a scaling factor of 10,000 followed by FindVariableFeatures using Variance Stabilization Transformation with the top 6000 features to be returned. The samples were scored on their cell cycle phases which would be used in the regression later. The FindIntegrationAnchors function using the CCA reduction method and IntegrateData was utilized to integrate the data together. The integrated data-set was then scaled in which mitochondrial percentage and cell cycle state was regressed out. A principal component analysis was performed and the top 15 dimensions were kept. Uniform Manifold and Projection (UMAP) was run on the dataset, in addition to FindNeighbors and FindClusters. Differential gene expression between clusters was performed using the FindMarkers function with the Wilocox test in Seurat. Violin plots and individual UMAP plots were all generated using the Seurat toolkit VlnPlot and FeaturePlot functions, respectively. Heatmaps were generated utilizing the pheatmap package (version 1.0.12).

### Single-nucleus RNA sequencing

Raw sequencing reads were demultiplexed to FASTQ format files using bcl2fastq (Illumina; version 2.20.0). Digital expression matrices were generated from the FASTQ files using Cell Ranger (*Zheng et al., 2017*; version 6.1.2) with the option to include intronic reads (--include-introns). Reads were aligned against the GRCm38 mouse genome assembly and gene counts were obtained, per-droplet, by summarizing exonic and intronic UMIs that overlapped with the GENCODE mouse annotation (release 24) for each gene symbol. In order to adjust for downstream effects of ambient RNA expression within mouse nuclei, we used the 'remove-background' module from CellBender (*Pita Juarez et al., 2022*; version 0.2.0) to remove counts due to ambient RNA molecules from the count matrices and to estimate the true cells. Genes were subsequently filtered such that only genes detected in two or more cells and with at least 6 total counts (across all cells) were retained. Sample demultiplexing via hashtag oligonucleotide sequences (HTOs) was performed with the Cumulus sc/snRNA-Seq processing pipeline (*Li et al., 2020*). Specifically, HTO quantification was performed with the Cumulus Tool on Feature Barcoding (*Li and Yang, 2024*), which provided a cell-by-HTO count matrix. This HTO count matrix, along with the gene count matrices generated via Cell Ranger (above) were used to assign each cell to their respective sample(s) with the demuxEM program. Only cells that were identified as singlets were retained (i.e. no cells identified as a multiplet or unassignable) in the per-sample CellBender-ed gene count matrices.

Cellbender output files were read into R (version 4.1.1) and processed utilizing the standard Seurat CCA and later RPCA integration workflows (version 4.3.0). Each of the hashed samples (24 in total) were merged with their respective pair to have a total of twelve samples consisting of six different groups. Each sample was filtered prior to downstream analysis based on their nCount_RNA, nFeature_RNA, and mitochondrial percentages. Samples were then normalized using a LogNormalization method with a scaling factor of 10000 followed by FindVariableFeatures using a Variance-Stabilizing Transformation as the method with the top 2000 features to be returned. The FindIntegrationAnchors function using the CCA reduction method and IntegrateData was utilized to integrate the data together. The integrated data-set was then scaled on which mitochondrial percentage was regressed. A principal component analysis was performed in which only the top 18 dimensions were retained. Uniform Manifold and Projection (UMAP), FindNeighbors, and FindClusters with a resolution of 0.4 was performed on the dataset. To remove doublets in the dataset, we used the package scDblFinder (1.8.0) and their function scDblFinder with the parameters of samples set to our twelve samples, dbr

set to NULL, dbr.sd set to 1, clusters set to FALSE, and multiSampleMode set to split. The object was then subsetted to only contain expected singlets. Differential gene expression between clusters was performed using the FindMarkers function with the Wilocox test in Seurat. Violin plots and individual UMAP plots were all generated using the Seurat toolkit VlnPlot and FeaturePlot functions, respectively. Heatmaps were generated utilizing the dittoSeq package (1.9.1) and pheatmap package (version 1.0.12).

After identifying the adipocyte population, we subsetted our object on that population, extracting the raw RNA counts on the cells for each of the six samples (YTN, OTN, Y3D, O3D, Y14D, O14D) (Y is young, O is 'Old' or as referred to in this paper, Aged). These samples were then integrated together using the standard RPCA integration workflow. There was no further filtering done on the reintegrated adipocyte population. Samples were normalized using a LogNormalization method with a scaling factor of 10000 followed by FindvariableFeatures using a Variance-Stabilizing Transformation as the method with the top 2000 features to be returned. The function SelectIntegrationFeatures was performed on the dataset where it was then scaled on which mitochondrial percentage was regressed, and principal components were found using the ScaleData and RunPCA functions. The FindIntegrationAnchors function using the ROCA reduction method and a k.anchors of 20 and IntegrateData was utilized to integrate the data together. After integration, the dataset was then scaled in which mitochondrial percentage was regressed on again. A principal component analysis was performed in which only the top 18 dimensions were retained. Uniform Manifold and Projection (UMAP), FindNeighbors, and FindClusters with a resolution of 0.2 was performed on the dataset. Differential gene expression between clusters was performed using the FindMarkers function with a Wilcoxon signed-rank test as the method in Seurat. Violin plots and individual UMAP plots were all generated using the Seurat toolkit VlnPlot and FeaturePlot functions, respectively. Heatmaps were generated utilizing the dittoSeq package (1.9.1) and pheatmap package (version 1.0.12).

Enrichment analysis was performed on the positively expressed genes with a $\log_2$ fold change (LFC) >0.25 and a $P_{adjusted}$ value <0.01 on comparison of the young 14 days cold and old 14 days cold groups in the DNL high cluster. The generated gene list, which was in order of significance, was fed into g:Profiler (version 0.2.1) using default parameters except with modifications to query as an ordered query against the 'mmusculus' database, a gSCS correction method for multiple testing, with domain scope set to annotated, and sources set to the Reactome database. The top six enriched pathways yielded from the database were taken and displayed in order of $P_{adjusted}$ value.

## Statistical methods

Mouse studies were performed with >n = 5 per group for p=0.05 with 95% power given the expected variability of examined phenotypes. Each experiment was independently replicated at least twice. Sample sizes are reported in figure legends. All bar graphs represent the mean ± SEM. A Student's t-test was used when two groups were compared. Where multiple conditions were compared, we applied two-way ANOVA with a Tukey correction for multiple comparisons. Only the Young vs. Aged comparisons were depicted on graphs for clarity, with additional multiple comparisons provided below. p Values are indicated by asterisks and defined as *p<0.05, **p<0.01 and ***p<0.001. All statistics were calculated with GraphPad Prism Version 10.0.3.

| Figure | Graph | Statistical test | Comparison | p value |
|--------|-------|------------------|------------|---------|
| 1B | *Ucp1* qPCR | Two-way ANOVA with a Tukey correction for multiple comparisons | 3D: Young vs. Aged | <0.001 |
| | | | 14D: Young vs. Aged | <0.001 |
| | | | Young: TN vs. 3D | <0.001 |
| | | | Young: TN vs. 14D | <0.001 |

*Continued on next page*

*Continued*

| Figure | Graph | Statistical test | Comparison | p value |
|--------|-------|------------------|------------|---------|
| 1B | *Cidea* qPCR | Two-way ANOVA with a Tukey correction for multiple comparisons | 3D: Young vs. Aged | <0.001 |
| | | | 14D: Young vs. Aged | <0.001 |
| | | | Young: TN vs. 3D | <0.001 |
| | | | Young: TN vs. 14D | <0.001 |
| | | | Young: 3D vs. 14D | 0.001 |
| 1B | *Dio2* qPCR | Two-way ANOVA with a Tukey correction for multiple comparisons | 3D: Young vs. Aged | <0.001 |
| | | | 14D: Young vs. Aged | 0.03 |
| | | | Young: TN vs. 3D | <0.001 |
| | | | Young: TN vs. 14D | 0.008 |
| | | | Young: 3D vs. 14D | <0.001 |
| 1B | *Ppargc1a* qPCR | Two-way ANOVA with a Tukey correction for multiple comparisons | 3D: Young vs. Aged | <0.001 |
| | | | Young: TN vs. 3D | <0.001 |
| | | | Young: TN vs. 14D | 0.03 |
| | | | Young: 3D vs. 14D | <0.001 |
| 2B | % tdTom%/Lin-;PDGFRa+ | Two-way ANOVA with an Uncorrected Fisher's LSD | Young: +/+vs. CER/+ | <0.001 |
| | | | Aged: +/+vs. CER/+ | <0.001 |
| 2B | % PDGFRa+/Lin- cells | Two-way ANOVA with an Uncorrected Fisher's LSD | Young: +/+vs. CER/+ | 0.008 |
| 4B | *Adipoq* qPCR | Two-way ANOVA with a Tukey correction for multiple comparisons | ICAM1: Young vs. Aged | <0.001 |
| | | | Young: DPP4 vs. CD142 | 0.006 |
| | | | Aged: DPP4 vs. ICAM1 | <0.001 |
| | | | Aged: DPP4 +vs. CD142 | 0.004 |
| 4B | *Fabp4* qPCR | Two-way ANOVA with a Tukey correction for multiple comparisons | ICAM1: Young vs. Aged | <0.001 |
| | | | Aged: DPP4 vs. ICAM1 | <0.001 |
| | | | Aged: ICAM1 vs. CD142 | 0.002 |
| 4D | *Adipoq* qPCR | Two-way ANOVA with a Tukey correction for multiple comparisons | Young: DPP4 vs. CD142 | 0.03 |
| | | | Young: ICAM1 vs. CD142 | 0.008 |
| | | | Aged: DPP4 vs. ICAM1 | 0.04 |
| | | | Aged: DPP4 vs. CD142 | <0.001 |
| | | | Aged: ICAM1 vs. CD142 | 0.006 |
| 4D | *Fabp4* qPCR | Two-way ANOVA with a Tukey correction for multiple comparisons | ICAM1: Young vs. Aged | 0.008 |
| | | | Young: DPP4 vs. ICAM1 | <0.001 |
| | | | Young: DPP4 vs. CD142 | <0.001 |
| | | | Aged: DPP4 vs. ICAM1 | <0.001 |
| | | | Aged: DPP4 +vs. CD142 | <0.001 |
| | | | Aged: ICAM1 vs. CD142 | 0.03 |
| 4F | *Adipoq* qPCR | Two-way ANOVA with an Uncorrected Fisher's LSD | Young: MIN vs. MAX | <0.001 |
| | | | Aged: MIN vs. MAX | <0.001 |

*Continued on next page*

*Continued*

| Figure | Graph | Statistical test | Comparison | p value |
|---|---|---|---|---|
| 4F | Fabp4 qPCR | Two-way ANOVA with an Uncorrected Fisher's LSD | Young: MIN vs. MAX | <0.001 |
| | | | Aged: MIN vs. MAX | <0.001 |
| 4G | *Ucp1* qPCR | Two-way ANOVA with a Tukey correction for multiple comparisons | DPP4: Young vs. Young +Iso | <0.001 |
| | | | DPP4: Aged vs. Aged +Iso | <0.001 |
| | | | ICAM1: Young vs. Young +Iso | <0.001 |
| | | | ICAM1: Aged vs. Aged +Iso | <0.001 |
| | | | ICAM1: Young +Iso vs. Aged +Iso | 0.02 |
| | | | CD142: Young vs. Young +Iso | 0.02 |
| | | | CD142: Aged vs. Aged +Iso | <0.001 |
| | | | Aged +Iso: Dpp4 +vs. Icam1+ | 0.002 |
| | | | Aged +Iso: Icam1 +vs. Cd142+ | 0.03 |
| 4H | *Ucp1* qPCR | Two-way ANOVA with a Tukey correction for multiple comparisons | MAX: Young vs. Young +Iso | <0.001 |
| | | | MAX: Aged vs. Aged +Iso | <0.001 |
| | | | Young +Iso: MIN vs. MAX | <0.001 |
| | | | Aged +Iso: MIN vs. MAX | <0.001 |
| 6E | *Npr3* qPCR | Two-way ANOVA with a Tukey correction for multiple comparisons | TN: Young vs. Aged | 0.001 |
| | | | 14D: Young vs. Aged | 0.01 |
| | | | Young: TN vs. 14D | 0.04 |
| | | | Aged: TN vs. 3D | 0.004 |
| | | | Aged: TN vs. 14D | 0.005 |
| S1A | Body mass | Two-way ANOVA with a Tukey correction for multiple comparisons | TN: Young vs. Aged | <0.001 |
| | | | 3D: Young vs. Aged | <0.001 |
| | | | 14D: Young vs. Aged | <0.001 |
| S1B | iWAT mass | Two-way ANOVA with a Tukey correction for multiple comparisons | TN: Young vs. Aged | 0.005 |
| | | | 3D: Young vs. Aged | 0.03 |
| S1B | iWAT mass % | Two-way ANOVA with a Tukey correction for multiple comparisons | No comparisons significant | N/A |
| S1E | *Ucp1* qPCR | Two-way ANOVA with an Uncorrected Fisher's LSD | Young: TN vs. 14D | 0.01 |
| | | | Aged: TN vs. 14D | 0.008 |
| S1E | *Cidea* qPCR | Two-way ANOVA with an Uncorrected Fisher's LSD | No comparisons significant | N/A |
| S4D | *Npr1* qPCR | Two-way ANOVA with a Tukey correction for multiple comparisons | Young: TN vs. 3D | 0.03 |

*Continued on next page*

*Continued*

| Figure | Graph | Statistical test | Comparison | p value |
|---|---|---|---|---|
| S4D | *Npr2* qPCR | Two-way ANOVA with a Tukey correction for multiple comparisons | No comparisons significant | N/A |
| S4H | *Acly* qPCR | Two-way ANOVA with a Tukey correction for multiple comparisons | 3D: Young vs. Aged | <0.001 |
| | | | 14D: Young vs. Aged | <0.001 |
| | | | Young: TN vs. 3D | <0.001 |
| | | | Young: TN vs. 14D | <0.001 |
| | | | Young: 3D vs. 14D | <0.001 |

## Acknowledgements

We thank members of the Seale lab for helpful advice and discussions. NIH grants DK120982 and DK121801 to PS; T32 HD083185 to CDH; T32 DK007314 to EF; RC2 DK116691 to EDR; P30-DK19525 (Penn Diabetes Research Center).

## Additional information

### Funding

| Funder | Grant reference number | Author |
|---|---|---|
| National Institutes of Health | DK120982 | Patrick Seale |
| National Institutes of Health | DK121801 | Patrick Seale |
| National Institutes of Health | T32 HD083185 | Corey D Holman |
| National Institutes of Health | T32 DK007314 | Ethan C Fein |
| National Institutes of Health | DK116691 | Evan D Rosen |
| National Institutes of Health | P30-DK19525 | Patrick Seale |

The funders had no role in study design, data collection and interpretation, or the decision to submit the work for publication.

### Author contributions

Corey D Holman, Conceptualization, Data curation, Formal analysis, Investigation, Methodology, Writing - original draft, Writing - review and editing; Alexander P Sakers, Conceptualization, Data curation, Formal analysis, Investigation; Ryan P Calhoun, Lan Cheng, Ethan C Fein, Data curation, Investigation; Christopher Jacobs, Data curation, Formal analysis, Investigation; Linus Tsai, Evan D Rosen, Data curation, Methodology; Patrick Seale, Conceptualization, Resources, Data curation, Formal analysis, Supervision, Funding acquisition, Investigation, Writing - original draft, Project administration, Writing - review and editing

### Author ORCIDs

Ethan C Fein ⓘ http://orcid.org/0000-0002-9798-9024
Linus Tsai ⓘ http://orcid.org/0000-0002-0134-6949
Patrick Seale ⓘ http://orcid.org/0000-0001-7119-1615

## Ethics

All animal procedures were approved and performed under the guidance of the University of Pennsylvania Institutional Animal Care and Use Committee (IACUC) (protocol #805649).

Reviewer #1 (Public review): https://doi.org/10.7554/eLife.87756.3.sa1
Reviewer #2 (Public review): https://doi.org/10.7554/eLife.87756.3.sa2
Author response https://doi.org/10.7554/eLife.87756.3.sa3

## Additional files

### Supplementary files
• MDAR checklist

### Data availability

scRNA-seq and snRNA-seq datasets are deposited in the Gene Expression Omnibus (GEO) under the superseries accession number GSE227441. Data analysis pipelines used for processing of raw sequencing data, integration and clustering can be obtained from: https://github.com/calhounr/Aging-impairs-cold-induced-beige-adipogenesis-and-adipocyte-metabolic-reprogramming (copy archived at *Calhounr, 2024*).

The following dataset was generated:

| Author(s) | Year | Dataset title | Dataset URL | Database and Identifier |
|---|---|---|---|---|
| Holman CD, Sakers AP, Calhoun RP, Cheng L | 2023 | Aging impairs cold-induced beige adipogenesis and adipocyte metabolic reprogramming | http://www.ncbi.nlm.nih.gov/geo/query/acc.cgi?acc=GSE227441 | NCBI Gene Expression Omnibus, GSE227441 |

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
