## [Editor Report · eLife assessment]

This **fundamental** study provides evidence that de novo beige adipogenesis from Pdgfra+ adipocyte progenitor cells is blocked during early aging in subcutaneous fat. The depth of the data at early ages is **compelling**, with rigorous cell tracing methodology employed. The study will aid in identifying new approaches to switch dormant adipocytes into an active thermogenic phenotype, and should be of interest to cell biologists at large.

---

## [Referee Report · Reviewer #1 (Public review)]

Thermogenic adipocyte activity associate with cardiometabolic health in humans, but decline with age. Identifying the underlying mechanisms of this decline is therefore highly important.

To address this task, Holman and co-authors present compelling data from their investigations of the effects of two major determinants of thermogenic activity: cold, which induce thermogenic de novo differentiation as well as conversion of dormant thermogenic inguinal adipocytes: and aging, which strongly reduce thermogenic activity. The authors study young and middle-aged mice at thermoneutrality and following cold exposure.

Using linage tracing, the authors conclude that the older group produce less thermogenic adipocytes from progenitor differentiation. However, they found no differences between thermogenic differentiation capacity between the age groups when progenitors are isolated and differentiated in vitro. This finding is consistent with previous findings in humans, demonstrating that progenitor cells derived from dormant perirenal brown fat of humans differentiate into thermogenic adipocytes in vitro. Taken together, this underscores that age-related changes in the microenvironment rather than autonomous alterations in the ASPCs explain the age related decline in thermogenic capacity, This is an important finding in terms of identifying new approaches to switch dormant adipocytes into an active thermogenic phenotype.

To gain insight into the age-related changes, the authors use single cell and single nuclei RNA sequencing mapping of their two age groups, comparing thermoneutral and cold conditions between the two groups. Interestingly, where the literature previously demonstrated that de novo lipogenesis (DNL) occurs in relation to thermogenic activation, the authors show that DNL in fact is activated in a white adipocyte cell type, whereas the beige thermogenic adipocytes form a separate cluster.

Considering recent findings, that adipose tissue contains several subtypes of ASPCs and adipocytes, mapping the changes at single cell resolution following cold intervention provides an important contribution to the field, in particular as an older group with limited thermogenic adaptation is analyzed in parallel with a younger, more responsive group. This model also allowed for detection of microenvironment as a determining factor of thermogenic response.

The use of only two time points (young and middle-aged) along the aging continuum limits the conclusions that can be made on aging as the only driver of the observed differences between the groups. Furthermore, as the authors also discuss, aging is a complex phenotype, and in this case the older mice were heavier and had larger fat depots, which should be taken into consideration when interpreting the data.

In conclusion, this study provides an important resource for further studies, which should investigate how the findings can be translated into humans for reactivation of dormant thermogenic fat and a potential improvement of metabolic health.

---

## [Referee Report · Reviewer #2 (Public review)]

This manuscript focused on why aging leads to decreased beiging of white adipose tissue. The authors used an inducible lineage tracing system and provided in vivo evidence that de novo beige adipogenesis from Pdgfra+ adipocyte progenitor cells is blocked during early aging in subcutaneous fat. Single-cell RNA sequencing of adipocyte progenitor cells and in vitro assays showed that these cells have similar beige adipogenic capacities in vitro. Single-cell nucleus RNA sequencing of mature adipocytes indicated that aged mice have more Npr3 high-expressing adipocytes in the subcutaneous fat from aged mice. Meanwhile, adipocytes from aged mice have significantly lower expression of genes involved in de novo lipogenesis, which may contribute to the declined beige adipogenesis.

The mechanism that leads to age-related impairment of white adipose tissue beiging is not very clear. The finding that Pdgfra+ adipocyte progenitor cells contribute to beige adipogenesis is novel and interesting. It is more intriguing that the aging process represses Pdgfra+ adipocyte progenitor cells from differentiating into beige adipocytes during cold stimulation. Mature adipocytes that have high de novo lipogenesis activity may support beige adipogenesis is also novel and worth further pursuing. The study was carried out with a nice experimental design, and the authors provided sufficient data to support the major conclusions. I only have a few comments that could potentially improve the manuscript.

(1) It is interesting that after three days of cold exposure, aged mice also have much fewer beige adipocytes. Is de novo adipogenesis involved at this early stage? Or does the previous beige adipocyte that acquired white morphology have a better "reactivation" in young mice? It would be nice if the author could discuss the possibilities.

(2) Is the absolute number of Pdgfra+ cells decreased in aged mice? It would be nice to include quantifications of the percentage of tomato+ beige adipocytes in total tomato+ cells to reflect the adipogenic rate.

---

## [Author Response]

The following is the authors’ response to the original reviews.

**Public Reviews**

We thank the reviewers for their insightful comments and helpful suggestions that allowed us to improve the manuscript.

**Reviewer #1:**
Thermogenic adipocyte activity associate with cardiometabolic health in humans but decline with age. Identifying the underlying mechanisms of this decline is therefore highly important.To address this task, Holman and co-authors investigated the effects of two major determinants of thermogenic activity: cold, which induce thermogenic de novo differentiation as well as conversion of dormant thermogenic inguinal adipocytes: and aging, which strongly reduce thermogenic activity. The authors study young and middle-aged mice at thermoneutrality and following cold exposure.Using linage tracing, the authors conclude that the older group produce less thermogenic adipocytes from progenitor differentiation. However, they found no differences between thermogenic differentiation capacity between the age groups when progenitors are isolated and differentiated in vitro. This finding is consistent with previous findings in humans, demonstrating that progenitor cells derived from dormant perirenal brown fat of humans differentiate into thermogenic adipocytes in vitro. Taken together, this underscores that age-related changes in the microenvironment rather than autonomous alterations in the ASPCs explain the age-related decline in thermogenic capacity. This is an important finding in terms of identifying new approaches to switch dormant adipocytes into an active thermogenic phenotype.To gain insight into the age-related changes, the authors use single cell and single nuclei RNA sequencing mapping of their two age groups, comparing thermoneutral and cold conditions between the two groups. Interestingly, where the literature previously demonstrated that de novo lipogenesis (DNL) occurs in relation to thermogenic activation, the authors show that DNL in fact is activated in a white adipocyte cell type, whereas the beige thermogenic adipocytes form a separate cluster.Considering recent findings, that adipose tissue contains several subtypes of ASPCs and adipocytes, mapping the changes at single cell resolution following cold intervention provides an important contribution to the field, in particular as an older group with limited thermogenic adaptation is analyzed in parallel with a younger, more responsive group. This model also allowed for detection of microenvironment as a determining factor of thermogenic response.The use of only two time points (young and middle-aged) along the aging continuum limits the conclusions that can be made on aging as the only driver of the observed differences between the groups. It should for example be noted that the older mice had higher weights and larger fat depots, thus the phenotype is complex and this should be taken into consideration when interpreting the data.In conclusion, this study provides an important resource for further studies on how to reactivate dormant thermogenic fat and potentially improve metabolic health.(1) The authors claim "Aging impairs cold-induced beige adipogenesis and adipocyte metabolic reprogramming". It is previously established in humans that aging strongly associate with a decline in thermogenic capacity. With this in mind, it is easy to accept that the reduced browning observed in the older group is due to age. However, the older group also have larger adipose depots, which also can be a confounding factor. I, therefore, recommend bringing this into the discussion and putting more focus on the complexity of the phenotype. For example, it could be discussed whether the de novo lipogenesis less due to that the adipocytes of older mice is already filled with more lipids. Additional time points along the aging continuum would be needed to make a strong conclusion about age as the determinant, but even so, aging is complex and further definitions and discussion would be needed.

We agree with the reviewer regarding the confounding effect of body weight changes. We have added a paragraph to the discussion (pasted below) to comment on the complexity of the phenotype and the contributing role of linked changes in body weight/composition.

“Aging is a complex process, and unsurprisingly, many pathways have been linked to the aging-related decline in beiging capacity. For example, increased adipose cell senescence, impaired mitochondrial function, elevated PDGF signaling and dysregulated immune cell activity during aging diminish beige fat formation (Benvie et al., 2023; Berry et al., 2017; Goldberg et al., 2021; Nguyen et al., 2021). Of note, older mice exhibit higher body and fat mass, which is associated with metabolic dysfunction and reduced beige fat development. While the effects of aging and altered body composition are difficult to separate, previous studies suggest that the beiging deficit in aged mice is not solely attributable to changes in body weight (Rogers et al., 2012). Further studies, including additional time points across the aging continuum may help clarify the role of aging and ascertain when beiging capacity decreases.”

(2) The study would gain from more comparisons to existing human studies and discussion on the translation potential of the findings. For example, how does the adipocyte subtypes identified in the current study translate to subtypes identified in human adipose tissue (e.g. Emont et al).

We analyzed the human adipose tissue atlas from Emont et al. 2022 (PMID: 35296864). We did not find any obvious homologous human adipocyte subtypes. However, this and other available human single cell studies have not investigated the effects of cold exposure on white adipose tissue depots, which may be necessary to reveal DNL-high and especially beige adipocytes.

(3) The group has contributed multiple studies demonstrating that Prdm16 is a major inducer of a thermogenic phenotype, and the literature shows that Prdm16 promote a thermogenic phenotype in favour of a fibrogenic aging phenotype. It would therefore be interesting to see how Prdm16 is regulated in the current data set, across adipocytes subtypes, age groups and temperature conditions.

We thank the reviewer for this comment. Previous studies showed that PRDM16 protein and not mRNA levels are downregulated during aging (Wang et al., 2019, Cell Metab, PMID: 31155495; Wang et al., 2022, Nature, PMID: 35978186). Consistent with this, we did not observe an agingassociated reduction in Prdm16 mRNA levels in adipocytes in our dataset. We did observe enrichment of Prdm16 mRNA levels in beige adipocytes relative to other adipocyte clusters. We included these data in Fig. 5F.

(4) In Figure 1, it is difficult to understand why the 6 weeks cold exposure is not shown in relation to the thermoneutrality, 3 days and 2-week cold exposure? It would be useful to have this in the same graph relating the levels and showing all four marker genes for all time points.

These experiments were done at different times using separate groups of mice. We have now clarified this in the figure legend.

(5) The older mice had larger inguinal fat depots, suggesting more lipids stored. The morphology of adipose tissue has previously been shown to be modulated by cold acclimation and is also the main similarity between brown adipose tissue in adult humans and young mice beige adipose tissue. Fig S2b suggests smaller adipocytes in the young group. It would also be useful, for comparison to published data, if authors show tissue sections with H&E of their model.

Good point. We added panels showing H&E staining of serial iWAT sections, showing changes in tissue morphology across age and temperature conditions (Figure S1F).

(6) The authors use t-tests to compare the differences induced by e.g. cold or min vs max cell culture media etc, within each age group. However, in my opinion, a two-way Anova with post-tests would be more informative as this would allow for testing the effects of the two age categories on any quantitative variable and allow for addressing whether there is an interaction between the categories.

Following the reviewer’s recommendation, we applied two-way ANOVA with a Tukey correction for multiple comparisons for categorical comparisons with different age groups and conditions. P values from all significant multiple comparison tests are now included within the methods section.

(7) In Figure 5F, please include Adipoq expression between clusters and please add a reference to why Nnat is considered a canonical white adipocyte marker.

We added Adipoq to the violin plot in Figure 5F, showing differential expression across adipocyte clusters. We included a line in the results section to highlight this observation:

“Interestingly, Adiponectin (Adipoq) was differentially expressed across adipocyte clusters, with higher levels in Npr3-high and DNL-high cells.”

We removed “canonical” and added references for Nnat and Lep as white marker genes.

(8) After 14 days of cold exposure, it looks like the DNL high population divides into two populations, did the authors explore if there was any differences between these clusters?

We also noticed this apparent division and explored this question. However, upon increasing the resolution for clustering and splitting the DNL high population, there were no obvious differentially expressed genes that defined the two subclusters. Thus, we opted to keep them together.

(9) As cold treatment transform a subset of cells, can authors perform a data-driven analysis to visualize the directions in their single nuclei data sets by using monocle pseudotime and/or velocity analyses?

This is a good question. We spent a long time trying to address this question using several trajectory and pseudotime analysis methods, including Velocity (scVelo), Slingshot and Dynoverse. Unfortunately, we were unable to obtain concordant results using at least two different methods and felt that the analyses were unreliable.

**Reviewer #2:**
This manuscript focused on why aging leads to decreased beiging of white adipose tissue. The authors used an inducible lineage tracing system and provided in vivo evidence that de novo beige adipogenesis from Pdgfra+ adipocyte progenitor cells is blocked during early aging in subcutaneous fat. Single-cell RNA sequencing of adipocyte progenitor cells and in vitro assays showed that these cells have similar beige adipogenic capacities in vitro. Single-cell nucleus RNA sequencing of mature adipocytes indicated that aged mice have more Npr3 high-expressing adipocytes in the subcutaneous fat from aged mice.

Meanwhile, adipocytes from aged mice have significantly lower expression of genes involved in de novo lipogenesis, which may contribute to the declined beige adipogenesis.

The mechanism that leads to age-related impairment of white adipose tissue beiging is not very clear. The finding that Pdgfra+ adipocyte progenitor cells contribute to beige adipogenesis is novel and interesting. It is more intriguing that the aging process represses Pdgfra+ adipocyte progenitor cells from differentiating into beige adipocytes during cold stimulation. Mature adipocytes that have high de novo lipogenesis activity may support beige adipogenesis is also novel and worth further pursuing. The study was carried out with a nice experimental design, and the authors provided sufficient data to support the major conclusions. I only have a few comments that could potentially improve the manuscript.(1) It is interesting that after three days of cold exposure, aged mice also have much fewer beige adipocytes. Is de novo adipogenesis involved at this early stage? Or does the previous beige adipocyte that acquired white morphology have a better "reactivation" in young mice? It would be nice if the author could discuss the possibilities.

This is a good question. We did not evaluate beige adipogenesis at the 3d timepoint. However, a previous study demonstrates that 3d of cold exposure is sufficient to promote de novo beige adipogenesis (Wang et al., Nat Med. 2013, PMID: 23995282). We observed that beige adipogenesis from Pdgfra+ cells are a relatively minor contributor to beige adipocyte development, even after long term cold exposure in young mice. Based on these data, we presume that beige adipocyte activation (or re-activation) is the dominant mechanism for beige adipocyte development.

To clarify this point, we have included the following lines in the manuscript:

“Previous studies in mice using an adipocyte fate tracking system show that a high proportion of beige adipocytes arise via the de novo differentiation of ASPCs as early as 3 days of cold (Wang et al., 2013).”

“Based on these findings, we presume that mature (dormant beige) adipocytes serve as the major source of beige adipocytes in our cold-exposure paradigm. However, long-term cold exposure also recruits smooth muscle cells to differentiate into beige adipocytes; a process that we did not investigate here (Berry et al., 2016; Long et al., 2014; McDonald et al., 2015; Shamsi et al., 2021).”

(2) Is the absolute number of Pdgfra+ cells decreased in aged mice? It would be nice to include quantifications of the percentage of tomato+ beige adipocytes in total tomato+ cells to reflect the adipogenic rate.

We presented FACS quantification of tdTomato+/Pdgfra+ cells in Fig. 2B. We added a graph showing the percentage of Pdgfra+ cells of total live, lin- cells in adipose tissue; this showed no difference between young and aged mice. We did not perform FACS quantification of tdTomato+ beige adipocytes due to the technical challenges with sorting adipocytes. Quantification of total tdTomato+ cells was also unreliable and inconsistent due to the widespread labeling of fibroblasts, blood vessels, along with traced adipocytes. Thus, we did not include this analysis.

(3) Line 112, the sentence seems to be not finished.

This has been corrected.